

# An improved air mass factor calculation for NO$_2$ measurements from GOME-2

Song Liu[1], Pieter Valks[1], Gaia Pinardi[2], Jian Xu[1], Athina Argyrouli[4,1], Ronny Lutz[1], L. Gijsbert Tilstra[3], Vincent Huijnen[3], François Hendrick[2], and Michel Van Roozendael[2]

[1]Deutsches Zentrum für Luft- und Raumfahrt (DLR), Institut für Methodik der Fernerkundung (IMF), Oberpfaffenhofen, Germany
[2]Belgian Institute for Space Aeronomy (BIRA-IASB), Brussels, Belgium
[3]Royal Netherlands Meteorological Institute (KNMI), De Bilt, the Netherlands
[4]Technical University of Munich (TUM), Department of Civil, Geo and Environmental Engineering, Chair of Remote Sensing Technology, Munich, Germany

*Correspondence to:* Song Liu (Song.Liu@dlr.de)

**Abstract.** An improved tropospheric nitrogen dioxide (NO$_2$) retrieval algorithm from the Global Ozone Monitoring Experiment-2 (GOME-2) instrument based on air mass factor (AMF) calculations performed with more realistic model parameters is presented. The viewing angle-dependency of surface albedo is taken into account by improving the GOME-2 Lambertian-equivalent reflectivity (LER) climatology with a directionally dependent LER (DLER) dataset over land and an
ocean surface albedo parametrization over water. A priori NO$_2$ profiles with higher spatial and temporal resolutions are obtained from the IFS(CB05BASCOE) chemistry transport model based on recent emission inventories. A more realistic cloud treatment is provided by a Cloud-As-Layers (CAL) approach, which treats the clouds as uniform layers of water droplets, instead of the current Clouds-as-Reflecting-Boundaries (CRB) model, which assumes the clouds as Lambertian reflectors. Improvements in the AMF calculation affect the tropospheric NO$_2$ columns on average within $\pm15\%$ in winter and $\pm5\%$ in summer over
largely polluted regions. In addition, the impact of aerosols on our tropospheric NO$_2$ retrieval is investigated by comparing the concurrent retrievals based on ground-based aerosol measurements (explicit aerosol correction) and aerosol-induced cloud parameters (implicit aerosol correction). Compared to the implicit aerosol correction through the CRB cloud parameters, the use of CAL reduces the AMF errors by more than 10%. Finally, to evaluate the improved GOME-2 tropospheric NO$_2$ columns, a validation is performed using ground-based Multi-AXis Differential Optical Absorption Spectroscopy (MAXDOAS) mea-
surements at the BIRA-IASB Xianghe station. The improved tropospheric NO$_2$ dataset shows good agreement with coincident ground-based measurements with a correlation coefficient of 0.94 and a relative difference of -9.9% on average.

## 1 Introduction

Tropospheric nitrogen dioxide (NO$_2$) is an important air pollutant that harms the human respiratory system even with short exposures (Gamble et al., 1987; Kampa and Castanas, 2008) and contributes to the formation of tropospheric ozone, urban haze,
and acid rain (Charlson and Ahlquist, 1969; Crutzen, 1970; McCormick, 2013). Besides the natural sources like soil emissions and lightning, the combustion-related emission sources from anthropogenic activities like fossil fuel consumption, car





traffic, and biomass burning produce substantial amounts of $NO_2$. Satellite measurements from the Global Ozone Monitoring Experiment (GOME) (Burrows et al., 1999), the SCanning Imaging Absorption SpectroMeter for Atmospheric CHartographY (SCIAMACHY) (Bovensmann et al., 1999), the Ozone Monitoring Instrument (OMI) (Levelt et al., 2006), and the Global Ozone Monitoring Experiment-2 (GOME-2) (Callies et al., 2000; Munro et al., 2016) have produced global $NO_2$ measure-

ments on long-time scales. In the following years, new generation instruments like the TROPOspheric Monitoring Instrument (TROPOMI) (Veefkind et al., 2012) aboard the Sentinel-5 Precursor satellite and geostationary missions like the Sentinel-4 (Ingmann et al., 2012) will deliver $NO_2$ dataset with high spatial resolution and short revisit time.

The GOME-2 instruments, which are on the EUMETSAT's MetOp-A and MetOp-B satellites (referred to as GOME-2A and GOME-2B in this work), have provided a long-term $NO_2$ dataset started in the year 2007. This dataset will be extended with

a third GOME-2 aboard the MetOp-C satellite, launched in December 2018. GOME-2 is a scanning spectrometer measuring the solar irradiance and Earth's backscattered radiance in the UV and VIS spectral ranges with a spectral resolution of 0.2-0.4 nm and a spatial resolution of 80 km×40 km (the spatial resolution has been increased to 40 km×40 km for GOME-2A from July 2013 onwards). GOME-2 provides morning observations of $NO_2$ at about 9:30 local time (LT), which complements early afternoon measurements e.g. from OMI or TROPOMI. The GOME-2 $NO_2$ measurements have been widely used in trend

studies, satellite dataset intercomparisons, and $NO_2$ emission estimations (e.g. Mijling et al., 2013; Hilboll et al., 2013, 2017; Krotkov et al., 2017; Irie et al., 2012; Gu et al., 2014; Miyazaki et al., 2017; Ding et al., 2017).

The $NO_2$ retrieval algorithm for the GOME-2 instrument contains three steps: the spectral fitting of the slant column (concentration along the effective light path) with the differential optical absorption spectroscopy (DOAS) method (Platt and Stutz, 2008) from the measured GOME-2 (ir)radiances, the separation of stratospheric and tropospheric contributions with a modi-

fied reference sector method, and the conversion of the tropospheric slant column to a vertical column with a tropospheric air mass factor (AMF) calculation. The quality of GOME-2 $NO_2$ measurements is strongly related to the calculation of the AMF, determined with a radiative transfer model, depending on a set of model parameters, such as viewing geometry, surface albedo, vertical distribution of $NO_2$, cloud, and aerosol. The model parameters, generally taken from external databases, contribute substantially to the overall AMF uncertainty, estimated to be in the range of 30-40% (Lorente et al., 2017).

The surface is normally assumed to be Lambertian with an isotropic diffuse reflection independent of viewing and illumination geometry in $NO_2$ retrieval (e.g. Boersma et al., 2011; van Geffen et al., 2019; Liu et al., 2019b). However, due to the occurrences of retroreflection and shading effects (mainly over rough surfaces like vegetation) and specular reflection (mainly over smooth surfaces like water), the Lambertian assumption is not always fulfilled. To account for the geometry-dependent surface scattering characteristics, the surface bidirectional reflectance distribution function (BRDF) (Nicodemus et al., 1992)

has been considered in previous studies (e.g. Zhou et al., 2010; Lin et al., 2014, 2015; Noguchi et al., 2014; Vasilkov et al., 2017; Lorente et al., 2018; Laughner et al., 2018; Qin et al., 2019), mainly based on measurements from MODerate resolution Imaging Spectroradiometer (MODIS) over land. However, due to the use of different instruments, biases are possibly introduced in the $NO_2$ retrieval. In addition, due to the generally unavailable full surface BRDF in all conditions and the complexity of accounting for BRDF, most of the current $NO_2$ and cloud retrievals still rely on Lambertian surface reflection (e.g., Boersma et al., 2018; van Geffen et al., 2019; Loyola et al., 2018; Desmons et al., 2019).



To account for the varying sensitivity of the satellite to $NO_2$ at different altitudes, a priori vertical profiles of $NO_2$ are required, generally prescribed using a chemistry transport model. The importance of the a priori $NO_2$ profiles used in the retrieval has been recognised earlier and motived the use of model data with high spatial resolution and/or high temporal resolution (e.g.

Valin et al., 2011; Heckel et al., 2011; Russell et al., 2011; McLinden et al., 2014; Yamaji et al., 2014; Kuhlmann et al., 2015; Lin et al., 2014; Boersma et al., 2016; Laughner et al., 2016). Within the Monitoring Atmospheric Composition and Climate (MACC) European project, a global data assimilation system for atmospheric composition forecasts and analyses has been developed and running operationally in Copernicus Atmosphere Monitoring Service (CAMS, http://atmosphere.copernicus.eu). The CAMS system relies on a combination of satellite observations with state-of-the-art atmospheric modelling (Flemming

et al., 2017), for which purpose the European Centre for Medium Range Weather Forecasts (ECMWF) numerical weather prediction Integrated Forecast System (IFS) was extended with modules for describing atmospheric composition (Flemming et al., 2015; Inness et al., 2015; Morcrette et al., 2009; Benedetti et al., 2009; Engelen et al., 2009; Agustí-Panareda et al., 2016). Profile forecasts from CAMS are planned to be applied in the operational $NO_2$ retrieval algorithm for the Sentinel-4 (Sanders et al., 2018) and Sentinel-5 (van Geffen et al., 2018) missions with the advantage of operational implementation and

high resolution. Lately, an advanced IFS system, referred to IFS(CB05BASCOE) (Huijnen et al., 2016, 2019) or IFS(CBA) for short, operates at high horizontal, vertical, and temporal resolutions based on recent emission inventories, providing an improved profile "representativeness".

Clouds influence the $NO_2$ retrieval through their increased reflectivity, their shielding effect on $NO_2$ column below the cloud, and multiple scattering that enhances absorption inside the cloud (Liu et al., 2004; Stammes et al., 2008; Kokhanovsky and

Rozanov, 2008). The presence of clouds is taken into account in the $NO_2$ AMF calculation using cloud parameters based on the Optical Cloud Recognition Algorithm (OCRA) and the Retrieval Of Cloud Information using Neural Networks (ROCINN) algorithms (Loyola et al., 2007, 2011). OCRA/ROCINN has been applied in the operational retrieval of trace gases from GOME (Van Roozendael et al., 2006), GOME-2 (Valks et al., 2011; Hao et al., 2014; Liu et al., 2019b), and TROPOMI (Heue et al., 2016; Theys et al., 2017; Loyola, in preparation). The latest version of OCRA/ROCINN (Lutz et al., 2016; Loyola et al., 2018)

provides two sets of cloud products: one treats clouds as ideal Lambertian reflectors in a "Clouds-as-Reflecting-Boundaries" (CRB) model, and the second treats clouds as uniform layers of water droplets in a "Clouds-As-Layers" (CAL) model. The CAL model, which allows for the penetration of photons through the cloud, is more realistic than the CRB model, which screens the atmosphere below the cloud (Rozanov and Kokhanovsky, 2004; Richter et al., 2015).

Aerosol scattering and absorption influence the top-of-atmosphere radiances and the light path distribution. The radiative

effect of scattering aerosols and clouds is comparable (i.e., the albedo effect, shielding effect, and multiple scattering), while the presence of absorbing aerosols generally reduces the sensitivity to $NO_2$ within and below the aerosol layer by decreasing the number of photons returning from this region to the satellite (Leitão et al., 2010). Because cloud retrieval does not distinguish between clouds and aerosols, the effect of aerosol on the AMF is normally corrected using an "implicit aerosol correction" by assuming that the effective clouds retrieved as Lambertian reflectors (i.e., using the CRB model) account for the effect of

aerosols on the light path (Boersma et al., 2004, 2011). Previous works have also applied an "explicit aerosol correction" for OMI pixels considering additional aerosol parameters (e.g. Lin et al., 2014, 2015; Kuhlmann et al., 2015; Castellanos et al.,



2015; Liu et al., 2019a; Chimot et al., 2019) and have reported large biases related to the implicit aerosol correction for polluted cases, likely because the simple CRB model can not fully describe the effects inherent to aerosol particles (Chimot et al., 2019).

The operational GOME-2 NO$_2$ products are generated with the GOME Data Processor (GDP) algorithm and provided by

DLR in the framework of EUMETSAT's Satellite Application Facility on Atmospheric Composition Monitoring (AC-SAF). The retrieval algorithm of total and tropospheric NO$_2$ from GOME-2 has been introduced by Valks et al. (2011, 2017) as implemented in the current operational GDP version 4.8. An updated slant column retrieval and stratosphere-troposphere separation have been presented by Liu et al. (2019b), and an improved AMF calculation is described in this paper, which will be implemented in the next version of GDP.

In the AC-SAF context (Hassinen et al., 2016), the NO$_2$ data derived from the GOME-2 GDP algorithm is being validated at BIRA-IASB by comparison with correlative observations from ground-based Multi-AXis Differential Optical Absorption Spectroscopy (MAXDOAS) (Pinardi et al., 2014, 2015; Pinardi, in preparation). The MAXDOAS instrument collects scattered sky light in a series of line-of-sight angular directions extending from the horizon to the zenith. High sensitivity towards absorbers near the surface is obtained for the smallest elevation angles, while measurements at higher elevations provide

information on the rest of the column. This technique allows the determination of vertically resolved abundances of atmospheric trace species in the lowermost troposphere (Hönninger et al., 2004; Wagner et al., 2004; Wittrock et al., 2004; Heckel et al., 2005).

In this work, we briefly introduce in Sect. 2 the reference retrieval algorithm for GOME-2 NO$_2$ measurements, which was described in detail in Liu et al. (2019b). We improve the AMF calculation in the reference retrieval algorithm in Sect. 3 by

accounting for the direction-dependency of surface albedo over land and over water, applying the advanced high-resolution IFS(CBA) a priori NO$_2$ profiles, and implementing the more realistic CAL cloud model. We investigate the properties of the implicit aerosol correction for aerosol-dominated scenes by comparing it to the explicit aerosol correction in Sect. 4. Finally, we show a validation of the GOME-2 tropospheric NO$_2$ columns using MAXDOAS datasets in Sect. 5.

## 2   Reference retrieval for GOME-2 NO$_2$ measurements

As described in Liu et al. (2019b), the NO$_2$ slant column retrieval applies an extended 425-497 nm wavelength fitting window (Richter et al., 2011) to include more NO$_2$ structures and an improved slit function treatment to compensate for the long-term and in-orbit drifts of the GOME-2 slit function. The uncertainty in the NO$_2$ slant columns is $\sim 4.4 \times 10^{14}$ molec/cm$^2$, calculated from the average slant column error using a statistical method (Valks et al., 2011, Sect. 6.1 therein). To determine the stratospheric NO$_2$ components, the STRatospheric Estimation Algorithm from Mainz (STREAM) method (Beirle et al., 2016)

with an improved treatment of polluted and cloudy pixels is adopted. The uncertainty in the GOME-2 stratospheric columns is $\sim 4$-$5 \times 10^{14}$ for polluted conditions based on the daily synthetic GOME-2 data and $\sim 1$-$2 \times 10^{14}$ for monthly averages.

Mainly focusing on the third retrieval step, we apply the tropospheric AMF $M$ conversion (Palmer et al., 2001; Boersma et al., 2004) to account for the average light path through the atmosphere:

$$M = \frac{\sum_l m_l(\boldsymbol{b}) x_l c_l}{\sum_l x_l} \tag{1}$$





**Table 1.** Ancillary parameters in deriving GOME-2 tropospheric $NO_2$ columns.

|  | reference retrieval (Liu et al., 2019b) | improved algorithm (this work) |
| --- | --- | --- |
| surface albedo | GOME-2 LER climatology | GOME-2 direction-dependent LER |
| a priori $NO_2$ profile | TM5-MP | IFS(CBA) |
| cloud parameter | OCRA/ROCINN_CRB | OCRA/ROCINN_CAL |

with $m_l$ the box-air mass factors (box-AMFs) in layer $l$, $x_l$ the partial columns from the a priori $NO_2$ profiles, and $c_l$ a correction coefficient to account for the temperature dependency of $NO_2$ cross-section (Boersma et al., 2004; Nüß et al., 2006).

The box-AMFs $m_l$ are derived using the multi-layered multiple scattering LIDORT (Spurr et al., 2001) radiative transfer model and stored in a look-up table (LUT) as a function of several model inputs $\boldsymbol{b}$, including GOME-2 viewing geometry, surface pressure, and surface albedo. Table 1 summarises the ancillary parameters used in the AMF calculation.

The surface albedo is described by a monthly Lambertian-equivalent reflectivity (LER) database (Tilstra et al., 2017), derived from GOME-2 measurements for the years 2007-2013 with a spatial resolution of $1.0°$ long$\times 1.0°$ lat for standard grid cells

and $0.25°$ long$\times 0.25°$ lat for coastlines (Tilstra et al., 2019). The LER is retrieved by matching the simulated reflectances to the Earth reflectance measurements for cloud-free scenes found with a statistic method (Koelemeijer et al., 2003; Kleipool et al., 2008; Tilstra et al., 2017).

The daily a priori $NO_2$ profiles are obtained from the three-dimensional chemistry transport model TM5-MP (Williams et al., 2017) with a horizontal resolution of $1°$ long$\times 1°$ lat for 34 vertical layers, as summarised in Table 2. The model is driven by

ECMWF ERA-Interim meteorological re-analysis (Dee et al., 2011) and updated every 3 h with interpolation of fields for the intermediate time periods. Compared to previous versions of TM model (e.g. Williams et al., 2009; Huijnen et al., 2010), which have been commonly used in tropospheric $NO_2$ retrieval studies (e.g. Boersma et al., 2011; Chimot et al., 2016; Lorente et al., 2017), the main advantages of TM5-MP is the better spatial resolution ($1°$ long$\times 1°$ lat), updated $NO_x$ emissions (year-specific MACCity emission inventory, Granier et al. (2011)), and improved chemistry scheme (an expanded version of the modified

CB05 chemistry scheme, Williams et al. (2013)).

In the presence of clouds, the AMF is derived based on the independent pixel approximation (Cahalan et al., 1994), which assumes the AMF as a linear combination of a cloudy-sky AMF $M_{cl}$ and a clear-sky AMF $M_{cr}$:

$$M = \omega M_{cl} + (1-\omega)M_{cr}, \tag{2}$$

where $\omega$ is the cloud radiance fraction. $M_{cl}$ is determined using Eq. (1) with the cloud surface regarded as a Lambertian reflector and with $m_l=0$ for layers below the cloud top pressure $c_p$. $\omega$ is derived from the GOME-2 cloud fraction $c_f$:

$$\omega = \frac{c_f I_{cl}}{(1-c_f)I_{cr} + c_f I_{cl}} \tag{3}$$

with $I_{cr}$ the backscattered radiance for a clear scene derived using LIDORT and $I_{cl}$ for a cloudy scene. Note that the cloud fraction $c_f$ is a radiometric or effective cloud fraction instead of a geometric one.





**Table 2.** Summary of chemistry transport model specifications.

|  | TM5-MP (Huijnen et al., 2010) (Williams et al., 2017) | IFS(CBA) (Flemming et al., 2015) (Huijnen et al., 2016) |
|---|---|---|
| horizontal resolution | 1° (long/lat) | ∼80 km (T255) or ∼0.7° (long/lat) |
| vertical resolution | 34 layers (∼6 layers below 1.5 km) | 137[1] layers (∼12 layers below 1.5 km) |
| temporal resolution | 2 h archiving | 1 h archiving |
| meteorological fields | ECMWF 3 h | ECMWF online (initialized with ERA-5) |
| tropospheric chemistry | modified CB05 (Williams et al., 2013) | modified CB05 (Williams et al., 2013) |
| anthropogenic emission | MACCity (Granier et al., 2011) | CAMS_GLOB_ANT v2.1 (Granier et al., 2019) |
| advection | slopes scheme (Russell and Lerner, 1981) | semi-Lagrangian scheme as described in Temperton et al. (2001) and Hortal (2002) |
| convection | ECMWF | Bechtold et al. (2014) |
| diffusion | Holtslag and Boville (1993) | Beljaars and Viterbo (1998) |

[1] 69 layers are employed in this study.

The GOME-2 cloud properties are derived by the OCRA and the ROCINN algorithms (Loyola et al., 2007, 2011; Lutz et al., 2016; Loyola et al., 2018). Since clouds generally have a higher reflectivity than the ground, OCRA calculates the radiometric cloud fractions by comparing the measured reflectances in 3 broadband wavelength regions across the UV-VIS-NIR region with corresponding cloud-free background composite maps using a RGB color space approach. The monthly cloud-free background map is calculated from GOME-2A measurements for the years 2008-2013, accounting for instrumental degradation and dependencies on viewing zenith angle (VZA), latitude, and season. With the radiometric cloud fractions from OCRA as input, ROCINN retrieves the cloud top pressures (cloud top heights) and cloud albedo (cloud optical depth) by comparing the simulated and measured satellite radiances in the $O_2$ A band around 760 nm using regularization theory. Based on the independent pixel approximation and the CRB cloud model, the ROCINN algorithm treats the clouds as Lambertian surfaces.

## 3 Improved AMF calculation

### 3.1 Surface albedo

The dependency of surface reflection on incoming and outgoing directions is mathematically described by the BRDF (Nicodemus et al., 1992), which shows a "hot spot" of increased reflectivity in backward scattering directions over rough surfaces like vegetation and a strong forward scattering peak near "sun glint" geometries over smooth surfaces like water. In this study, we account for the direction-dependency of surface albedo for the GOME-2 LER climatology by applying a directionally depen-





dent LER (DLER) dataset over land surfaces (see Sect. 3.1.1) and by implementing an ocean surface albedo parametrization over water surfaces (see Sect. 3.1.2).

### 3.1.1 Over land

To account for the surface BRDF in our $NO_2$ AMF calculation over land, the surface reflectivity is described by a GOME-2
DLER dataset (Tilstra et al., 2019) that captures the VZA-dependency. Compared to the traditional GOME-2 LER climatology (Tilstra et al., 2017), derived from a range of viewing angles ($\sim$115° for GOME-2 measurements covering the directions from east to west), the GOME-2 DLER dataset is derived by dividing the range of viewing angles into five segments and applying the same retrieval method as in the traditional GOME-2 LER determination for each segment with a parabolic fit to parametrize the VZA-dependency. The main idea of this VZA-dependency parametrization is to use the VZA as a proxy of observation
geometry over land, since solar zenith angle (SZA) and relative azimuth angle (RAA) are nearly constant at a given latitude and thus have been captured in the original GOME-2 LER dataset.

For each GOME-2 measurement, the surface DLER $\alpha_{DLER}$ is calculated as:

$$\alpha_{DLER} = \alpha_{LER} + c_0 + c_1 \times \theta + c_2 \times \theta^2 \tag{4}$$

with VZA $\theta$ positive on the west side of the orbit swath and negative on the east side of the orbit swath. $c_0$, $c_1$, and $c_2$
are parabolic fitting coefficients depending on latitude, longitude, month, and wavelength. The non-directional LER $\alpha_{LER}$ is taken from the traditional GOME-2 LER climatology. Note that no directionality is provided by the DLER dataset over water (without sea ice cover), mainly due to the dependency on parameters such as wind speed and chlorophyll concentration, which can not be cast into climatology easily. Additionally, due to the strong solar and viewing angles-dependency of specular reflection, changes of the solar position during a month influence the albedo over water bodies much more than for land, and
this influence is modelled and described in Sect. 3.1.2.

Figure 1a-c shows the traditional GOME-2 LER climatology, the GOME-2 DLER dataset over land, and their differences on 3 February and 5 August 2010. The DLER data shows a stronger increase for western viewing direction by $\sim$0.02 over vegetation, $\sim$0.05 over desert, and $\sim$0.2 over snow and ice, due to the increasing BRDF in the backward scattering direction. A slight change by up to 0.01 is found over vegetation and desert with enhancement for the central part of the orbit swath and
reduction for the east side of the orbit swath, and this effect is larger over snow and ice, resulted from the forward scattering peak or double scattering peak in the BRDF pattern for snow (Dumont et al., 2010). The difference in surface albedo is generally larger in winter, due to the change of surface condition and/or sun elevation, at the exception of desert.

Figure 2 compares the surface LER and DLER as a function of VZA and presents the impact on the clear-sky AMFs over western Europe (44° N-53° N, 0° E-7° E) and eastern China (21° N-41° N, 110° E-122° E) in February 2010. The surface albedo on the west side of the orbit swath (backward scattering direction) is higher for both regions by up to 0.024, which
reduces the calculated clear-sky AMFs by 9-14%. Smaller differences are found for the central and eastern viewing direction by up to 0.006 for surface albedo and up to 4% for clear-sky AMFs.





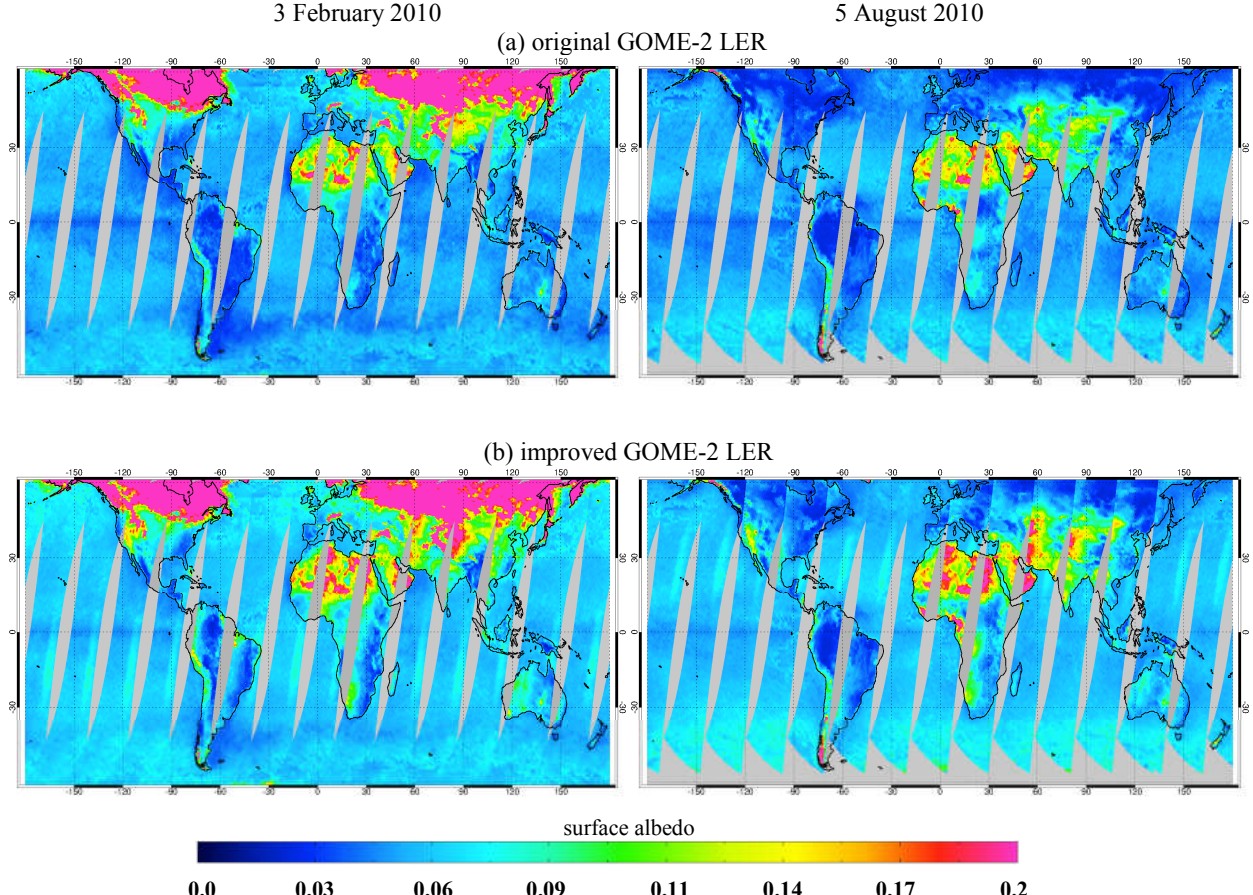

**Figure 1.** Map of GOME-2 surface LER climatology (Tilstra et al., 2017) version 3.1 in February and August (a), improved GOME-2 surface LER data taking into account the direction-dependency on 3 February and 5 August 2010 (b), and their differences over land (c) and over water (d) (figure continued on next page). The improvements are described in Sect. 3.1.1 for land and in Sect. 3.1.2 for water.

Figure 3 shows the differences in tropospheric $NO_2$ columns retrieved using surface LER and DLER dataset for a given day and for the monthly average in February and August 2010. The daily differences in tropospheric $NO_2$ columns are consistent with Fig. 1c, with a larger impact found over polluted regions. Taking Spain on 3 February 2010 as an example, the smaller surface DLER on the central part of the orbit swath by $\sim$0.005 results in a lower sensitivity to tropospheric $NO_2$ columns in the AMF calculation, and therefore the AMF decreases and the tropospheric $NO_2$ columns increases by $\sim 1 \times 10^{14}$ molec/cm$^2$ (3%). Vice versa, the surface DLER is higher by $\sim$0.02 on the west side of the orbit swath over eastern China at the same day, and thus the tropospheric $NO_2$ column is lower by $\sim 3 \times 10^{15}$ molec/cm$^2$ (11%). The monthly differences in tropospheric $NO_2$ columns show a larger reduction in winter by more than $5 \times 10^{14}$ molec/cm$^2$ over e.g. central Europe, South Africa, India, and eastern China, and by $\sim 1 \times 10^{14}$ molec/cm$^2$ over e.g. the eastern US, Southeast Asia, and Mexico.



3 February 2010                    5 August 2010

(c) difference over land

surface albedo (improved − original)

-0.02     -0.01     0.00     0.01     0.03     0.04     0.05     0.06

(d) difference over water

surface albedo (improved − original)

0.00     0.003     0.006     0.009     0.011     0.014     0.017     0.02

**Figure 1.** (figure continued from previous page)

The above results are in agreement with studies applying the BRDF product from MODIS to describe the dependency of land surface reflectance on illumination and viewing geometry (e.g. Zhou et al., 2010; Noguchi et al., 2014; Vasilkov et al., 2017; Lorente et al., 2018; Laughner et al., 2018; Qin et al., 2019). With a good agreement with the established MODIS BRDF product (Tilstra et al., 2019), the GOME-2 DLER dataset is derived from measurements of the instrument itself, consistent with the GOME-2 $NO_2$ observations, considering the illumination conditions, observation geometry, and instrumental characteristics, and therefore the use of GOME-2 DLER introduces no additional bias caused by the instrumental differences.

### 3.1.2 Over water

The surface reflectivity over water is described with an improved GOME-2 LER data using an ocean surface albedo parametrization (Jin et al., 2004, 2011) to account for the direction-dependency. Based on atmospheric radiation measurements



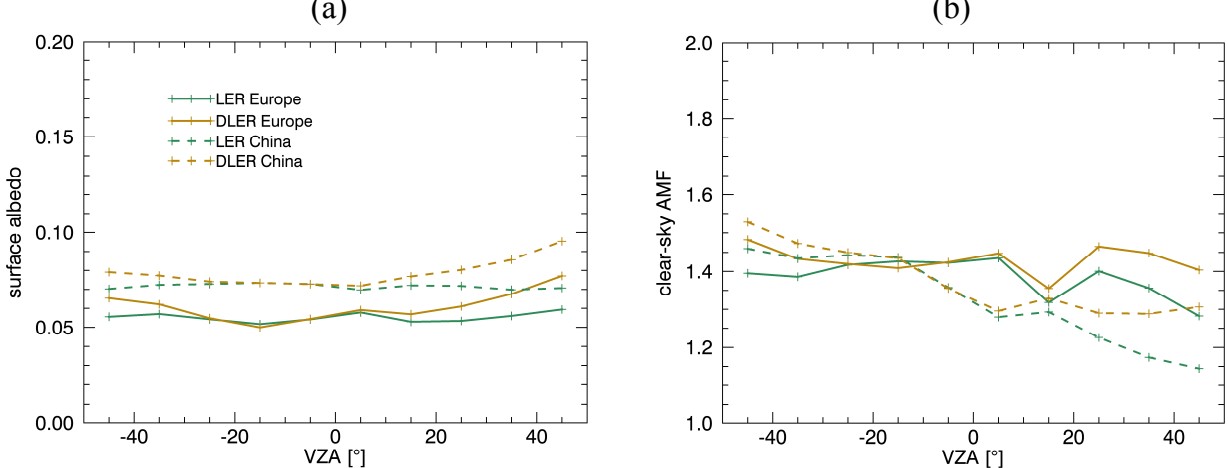

**Figure 2.** Comparison of GOME-2 LER climatology (Tilstra et al., 2017) and GOME-2 DLER data (a) and the impact on the clear-sky AMFs (b) over western Europe (44° N-53° N, 0° E-7° E) and eastern China (21° N-41° N, 110° E-122° E) as a function of VZA in February 2010 (VZAs are negative for observations on the east side of the orbit swath).

5 and the Coupled Ocean-Atmosphere Radiative Transfer (COART) model (Jin et al., 2006), the parametrization developed by Jin et al. (2011) derives the surface reflectivity for the direct and diffuse incident radiation separately and further divides each of them into contributions from surface and water, respectively. This parametrization has been used to derive ocean surface albedo (e.g. Séférian et al., 2018) and to generate satellite $NO_2$ product (e.g. Laughner et al., 2018).

Following Jin et al. (2011), the ocean surface albedo $\alpha_{total}$ is defined as:

$$\alpha_{total} = f_{dir}(\alpha_{dir}^s + \alpha_{dir}^w) + f_{dif}(\alpha_{dif}^s + \alpha_{dif}^w) \tag{5}$$

with $\alpha_{dir}^s$ and $\alpha_{dif}^s$ the direct and diffuse contribution of the surface reflection and $\alpha_{dir}^w$ and $\alpha_{dif}^w$ the direct and diffuse contribution of the volume scattering of water below the surface, respectively. The direct and diffuse fraction of downward surface flux $f_{dir}$ and $f_{dif}$ ($f_{dif} = 1 - f_{dir}$) are calculated using the online COART (https://satcorps.larc.nasa.gov/jin/coart.html). The direct surface albedo $\alpha_{dir}^s$, which is one main component of the total ocean surface albedo, describes the contribution

15 of Fresnel reflection depending on the incident angle, refractive index of seawater (1.343 at 460 nm), and slope distribution of the ocean surface (defined by Cox and Munk (1954) and related to wind speed (5 m/s from the climatological mean)). The diffuse surface albedo $\alpha_{dif}^s$ is difficult to formulate analytically due to its variation with atmospheric conditions and thus parametrized practically to be 0.06 for an assumed 5 m/s wind speed. The direct water volume albedo $\alpha_{dir}^w$ is considered for the case 1 waters (consist 99% of the ocean) and primarily affected by the chlorophyll concentration (0.2 mg/m³ from the global ocean average). The diffuse water volume albedo $\alpha_{dif}^w$ is defined by the $\alpha_{dir}^w$ at an effective incident direction (i.e. arccos(0.676)) and calculated to be 0.0145. The direct fraction of downward surface flux $f_{dir}$ is calculated with radiative

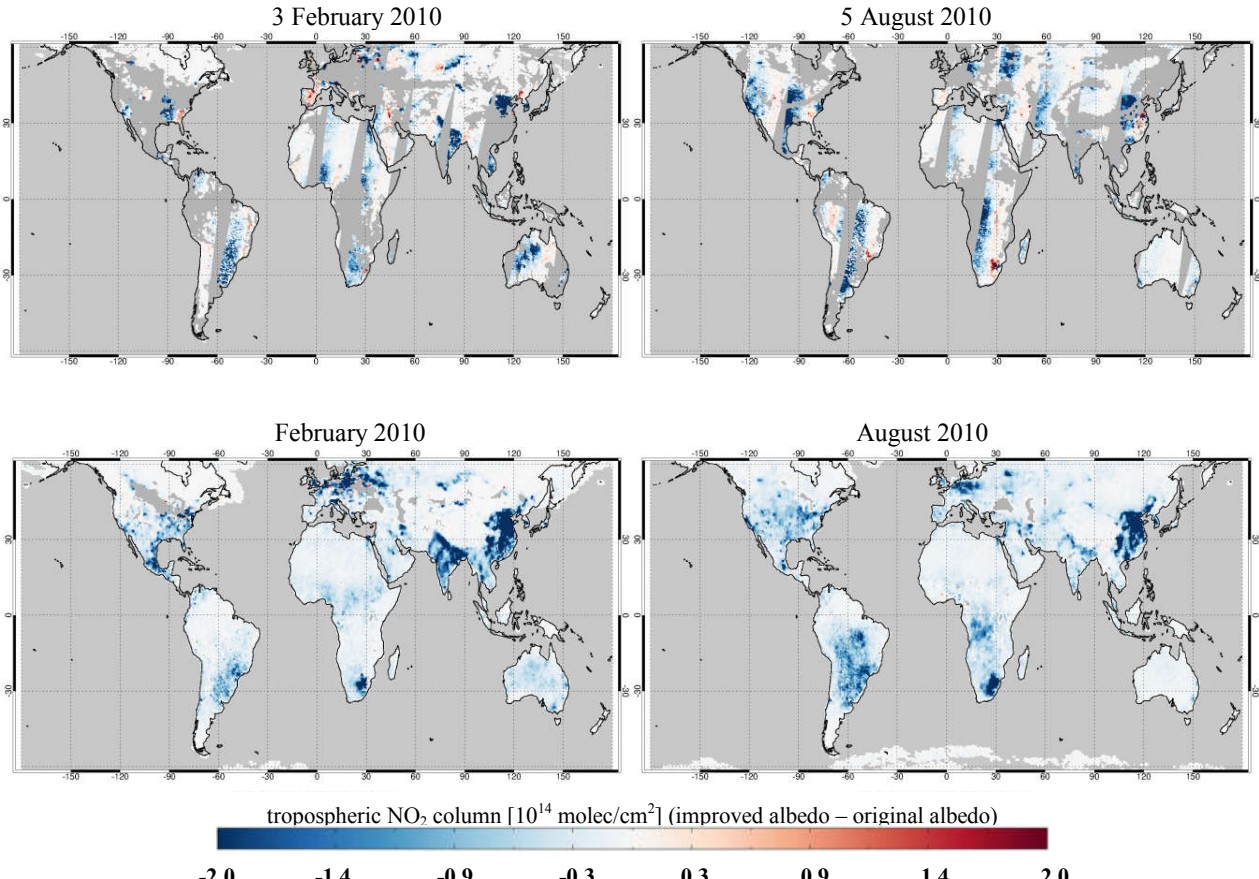

**Figure 3.** Differences in GOME-2 tropospheric NO$_2$ columns retrieved using GOME-2 LER and DLER dataset for a given day and for the monthly average in February and August 2010. Only measurements with cloud radiance fraction < 0.5 are included.

transfer simulation using a mid-latitude summer atmosphere with a marine aerosol optical depth of 1 (at 550 nm) and using a 100 m depth ocean with the average Petzold phase function for ocean particle scattering.

Figure 4 shows the parametrized ocean surface albedo for a non-glint situation and its four albedo components as a function of incident angle. The overall shape of the total ocean surface albedo $\alpha_{total}$ is dependent on incident angle with a peak near 70°, similar with Jin et al. (2004) and Laughner et al. (2018). The surface component ($\alpha^s_{dir} + \alpha^s_{dif}$) is larger than the water volume component ($\alpha^w_{dir} + \alpha^w_{dif}$), particularly for larger incident angles. The direct component ($\alpha^s_{dir} + \alpha^w_{dir}$) increases with incident angle with lower values than the diffuse component ($\alpha^s_{dif} + \alpha^w_{dif}$) for smaller incident angles (below 55°) and higher values for larger incident angles. The relative contribution of diffuse component to the total ocean surface albedo $f_{dif}$ increases from ∼0.65 to ∼1 with incident angle. It is worth noting that the four albedo components are independent of each other and thus flexible to update or replace.





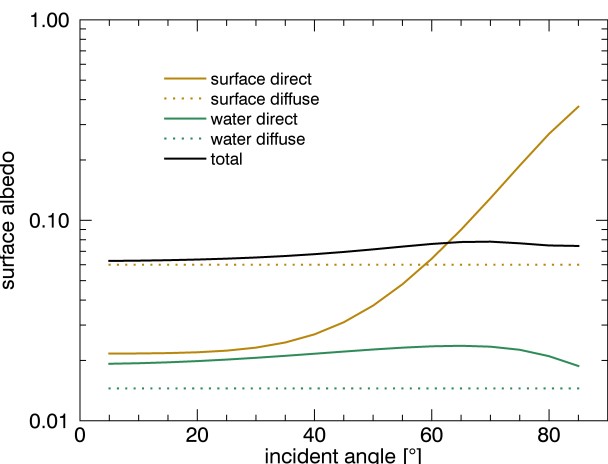

**Figure 4.** Parametrized ocean surface albedo for a non-glint condition and its albedo components due to direct and diffuse surface reflection and direct and diffuse water volume scattering as a function of incident angle.

Based on measurements over a long period (2007-2018 for version 3.1), the GOME-2 LER climatology provides mainly the diffuse component ($\alpha^s_{dif} + \alpha^w_{dif}$) over water bodies with minimized impact of direct contribution. Therefore, we replace the simplified expression of $\alpha^s_{dif} + \alpha^w_{dif}$ in Jin et al. (2011) with values taken from the GOME-2 LER climatology. This scheme enables the consideration of the direction-dependency for the GOME-2 LER climatology over water with minimal bias introduced. In addition, most of the ocean surface albedo studies (e.g. Ohlmann, 2003; Jin et al., 2004; Li et al., 2006; Jin et al., 2011; Laughner et al., 2018) employ a straightforward assumption that SZA is the only directional parameter involved in the parametrization, namely the incident angle is assumed to be equivalent to the SZA in the Fresnel reflection calculation. In this work, we apply the full equation to derive the local incident angle with dependencies on VZA and RAA also taken into account, and we additionally implement the Cox-Munk sun glitter model over glint-contaminated regions. See Cox and Munk (1954) and Gordon (1997) for more details on configuration and derivation.

Figure 1b,d presents the calculated ocean surface albedo and the differences with values taken from GOME-2 LER climatology on 3 February and 5 August 2010. Consistent with Vasilkov et al. (2017), the improved ocean surface albedo shows higher values by up to 0.015 at larger SZAs and VZAs, where the higher incident angles result in stronger Fresnel reflections, and by up to 0.025 over areas affected by sun glint, typically the eastern swath of GOME-2 orbits.

Figure 5 shows the impact of using updated ocean surface albedo on our GOME-2 $NO_2$ retrieval for a given day and for the monthly average in February and August 2010. The tropospheric $NO_2$ columns are reduced mainly over the polluted coastal regions with large $NO_2$ concentrations and with large SZAs and VZAs. For instance, the ocean surface albedo around Spain increases by $\sim$0.01 on 3 February 2010, leading to a decrease of tropospheric $NO_2$ columns by up to $8 \times 10^{14}$ molec/cm$^2$ (9%). The monthly average of tropospheric $NO_2$ columns decreases in winter by more than $3 \times 10^{14}$ molec/cm$^2$ near the coastal



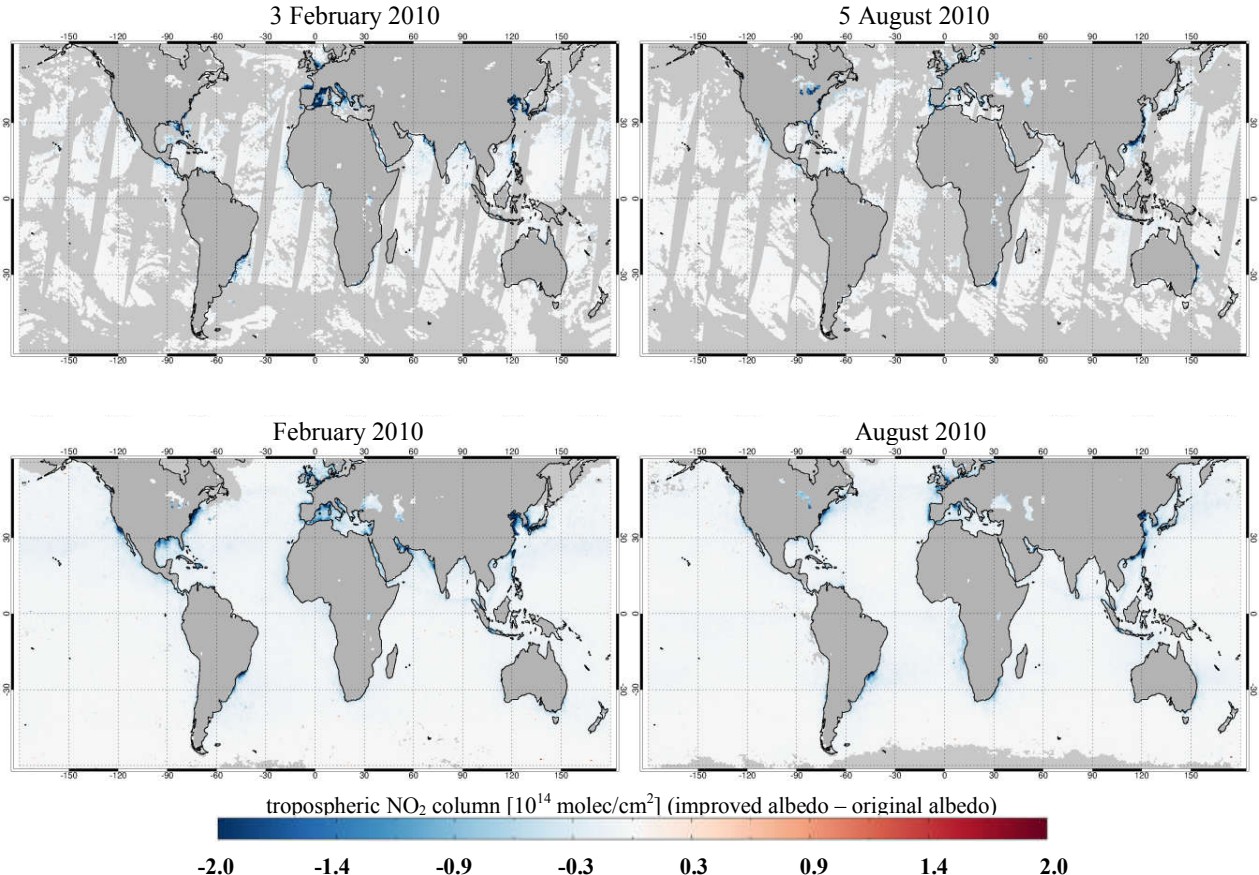

**Figure 5.** Differences in GOME-2 tropospheric NO₂ columns retrieved using the original GOME-2 LER climatology and the GOME-2 LER data improved with the ocean surface albedo parametrization for a given day and for the monthly average in February and August 2010. Only measurements with cloud radiance fraction < 0.5 are included.

5 area, e.g. around the US, eastern China, and Brazil, and by up to $1 \times 10^{14}$ molec/cm$^2$ along the shipping lanes, e.g. in the Mediterranean Sea, the Red Sea, and the maritime Southeast Asia.

## 3.2 A priori NO₂ profile

In regions with strong gradients in NO$_x$ emission in space and time, the significant variation of surface NO₂ can only be captured in a model with sufficient horizontal, vertical, and temporal resolutions. The advanced IFS(CBA) (Huijnen et al., 2016, 2019) global chemistry forecast and analysis system combines the stratospheric chemistry scheme developed for the Belgian Assimilation System for Chemical ObsErvations (BASCOE, Skachko et al. (2016)) and the modified CB05 tropo-

spheric chemistry scheme (Williams et al., 2013). As summarized in Table 2, the spatial resolution of IFS(CBA) is a reduced Gaussian grid at a spectral truncation of T255, which is equivalent to a grid spacing of ∼80 km globally (∼0.7°). The model



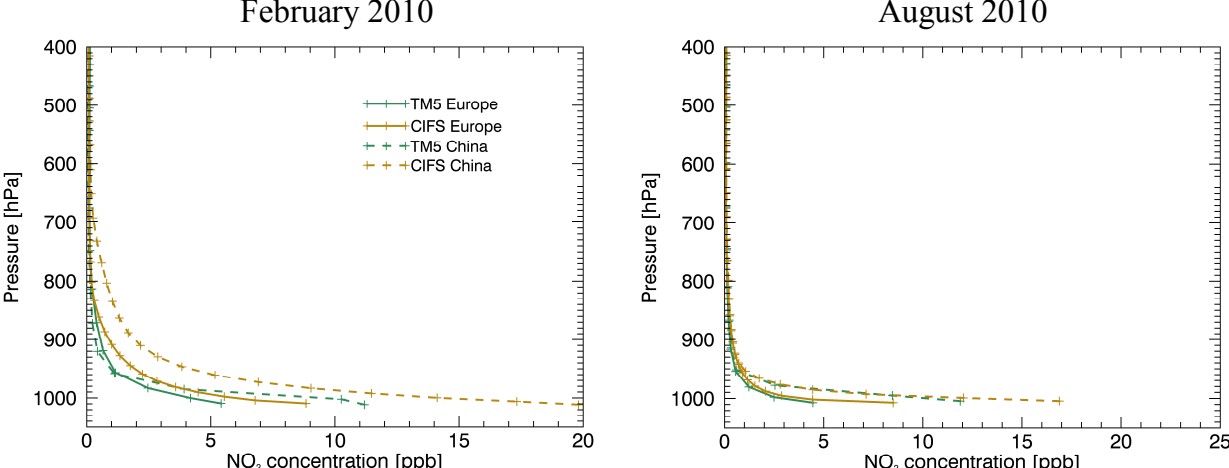

**Figure 6.** Area-averaged and monthly-averaged profiles from TM5-MP and IFS(CBA) at GOME-2 overpass time (9:30 LT) over western Europe (44° N-53° N, 0° E-7° E) and eastern China (21° N-41° N, 110° E-122° E) in February and August 2010.

is run with the standard 137 hybrid sigma-pressure layers as also used operationally in the ECMWF' forecast and reanalysis model. From this we select a vertical discretisation based on 69 vertical levels up to 0.1 hPa with ~12 layers in the boundary layer for further processing. An essential difference compared to TM5-MP is that in IFS(CBA) the chemistry is an integral part

of the meteorological forecast model. Here we use the forecast model from cycle 45r2, which is daily initialized using ERA5 meteorology. Additionally, anthropogenic emissions are based on the recently prepared CAMS_GLOB_ANT v2.1 emission inventory (Granier et al., 2019), while day-specific biomass burning emissions are taken from GFASv2.1 (Kaiser et al., 2012). The $NO_2$ data is available on hourly basis, based on which the profiles at the satellite measurement time can be obtained with a linear interpolation.

Figure 6 shows an intercomparison of area-averaged and monthly-averaged profiles from TM5-MP and IFS(CBA) at GOME-2 overpass time (9:30 LT) over western Europe (44° N-53° N, 0° E-7° E) and eastern China (21° N-41° N, 110° E-122° E) in February and August 2010. Generally, TM5-MP and IFS(CBA) show similar mean profile shapes over the two regions. In February, IFS(CBA) shows a larger boundary layer concentration and a sharper transition to the free troposphere over western Europe and a larger $NO_2$ gradients in the free troposphere over eastern China. In August, the $NO_2$ concentrations in the free

troposphere are lower than in February for both models due to the reduced emissions and the reduced lifetime of $NO_2$, and a larger surface layer $NO_2$ gradient is found for the IFS(CBA) model for both regions.

Figure 7 shows the daily TM5-MP and IFS(CBA) a priori $NO_2$ profiles over the Netherlands (52.8° N, 4.7° E) and China (39.1° N, 118.0° E) on 3 February 2010 as examples. IFS(CBA) shows a higher surface layer $NO_2$ concentration (more steep profile shape) and yields a tropospheric AMF reduced by 0.21 over the Netherlands, which will enhance the retrieved

tropospheric $NO_2$ column. In contrast, the tropospheric AMF increases by 0.06 over China due to the larger $NO_2$ gradients in the free troposphere (less steep profile shape) modelled by IFS(CBA).





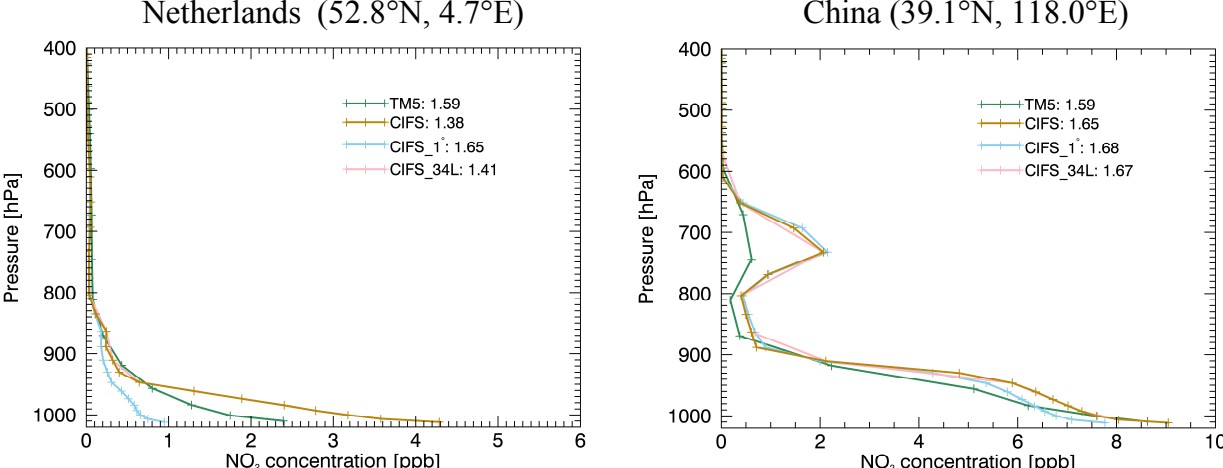

**Figure 7.** A priori $NO_2$ profiles from TM5-MP, IFS(CBA) (original resolution), and IFS(CBA) with different model resolutions over the Netherlands (52.8° N, 4.7° E) and China (39.1° N, 118.0° E) on 3 February 2010. The IFS(CBA) profiles are compared for 1° grid and for 34 layers. The calculated clear-sky tropospheric AMF is given next to each label.

Figure 8 shows the differences in tropospheric $NO_2$ columns retrieved using TM5-MP and IFS(CBA) a priori $NO_2$ profiles for a given day and for the monthly average in February and August 2010. The differences are consistent with the changes in the profile shapes in Fig. 6 and 7. The use of IFS(CBA) generally increases the tropospheric $NO_2$ columns over polluted regions by up to $2 \times 10^{15}$ molec/cm$^2$, e.g. over western Europe, eastern US, and Argentina, and decreases the values by up to $1 \times 10^{15}$ molec/cm$^2$, e.g. over central Africa, South Africa, and Brazil. In February, however, a strong enhancement by $\sim 7 \times 10^{15}$ molec/cm$^2$ is found over northern Germany and Poland, and a strong reduction by $\sim 4 \times 10^{15}$ molec/cm$^2$ is found over the North China Plain. The differences in Fig. 8 are likely related to the different chemical mechanism, transport scheme, and emission inventories employed by the model as well as the different model resolutions.

To quantify the effect of model resolutions, a more detailed analysis for IFS(CBA) is implemented with 1° grids for horizontal resolution, with 34 layers for vertical resolution, and with 2-hours time steps for temporal resolution, respectively. These values are of the same order of magnitude as the model resolutions of TM5-MP and other chemistry transport models currently employed in the satellite retrieval of $NO_2$ (e.g. van Geffen et al., 2019; Lorente et al., 2017; Boersma et al., 2018; Liu et al., 2019b). Figure 7 compares the IFS(CBA) a priori $NO_2$ profiles with original and different model resolutions over the Netherlands (52.8° N, 4.7° E) and China (39.1° N, 118.0° E) on 3 February 2010. Both examples are located at polluted coastal regions, which typically have a large heterogeneity and variability in the $NO_2$ distribution. The AMFs differ by more than 0.02 for both examples due to differences in horizontal and vertical resolutions. The current 2-hours temporal sampling and subsequent linear interpolation between the sampling points is sufficient for the retrieval of tropospheric $NO_2$ columns (not shown). When a coarser spatial resolution is used, the "domain-averaged" profiles generally show an increased surface $NO_2$ concentration for unpolluted domain and the opposite for emission source. Consequently, the AMF is underestimated

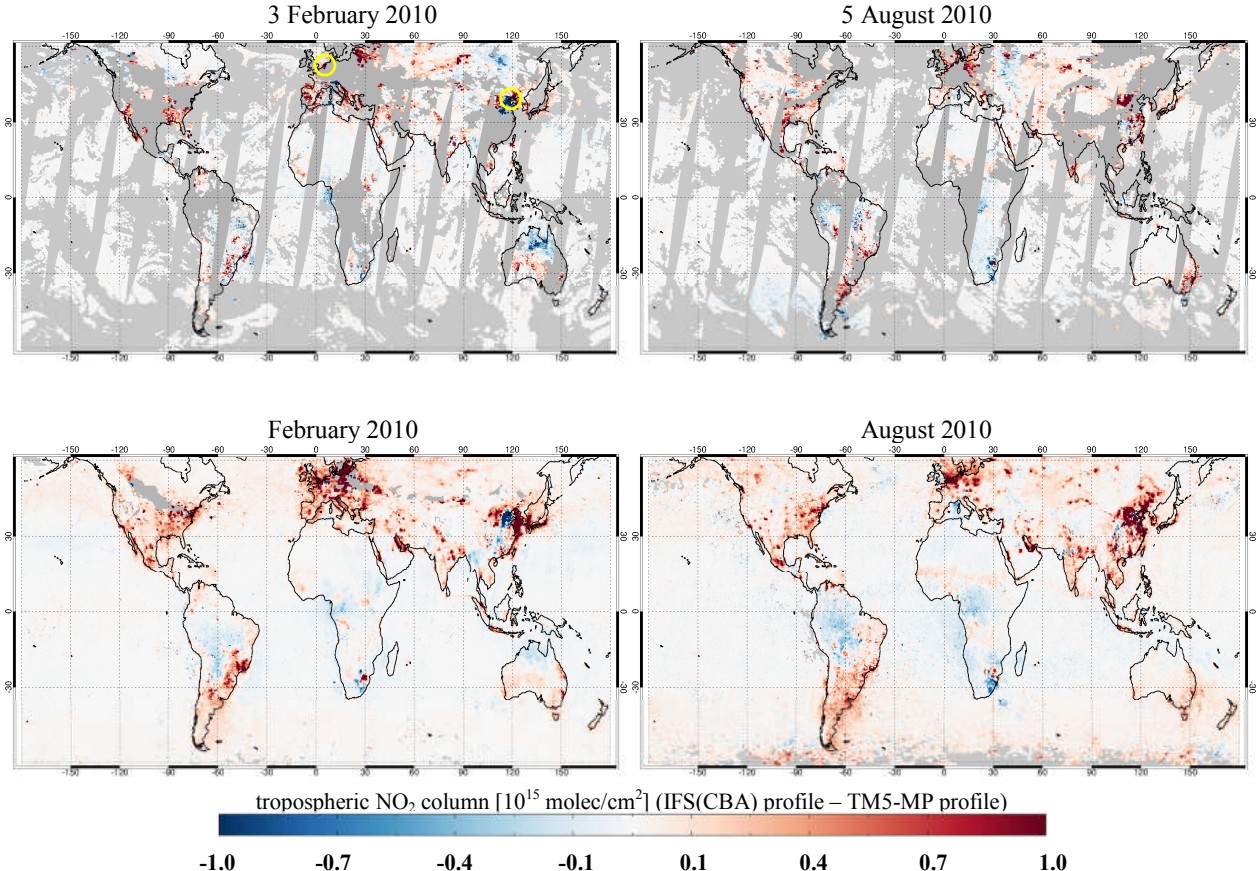

tropospheric NO$_2$ column [$10^{15}$ molec/cm$^2$] (IFS(CBA) profile − TM5-MP profile)

**Figure 8.** Differences in tropospheric NO$_2$ columns retrieved using TM5-MP and IFS(CBA) a priori NO$_2$ profiles for a given day and for the monthly average in February and August 2010. Yellow circles on 3 February 2010 indicate locations in Fig. 7. Only measurements with cloud radiance fraction < 0.5 are included.

for unpolluted areas and overestimated for polluted areas. When the number of layers is reduced, the coarser sampling points can not capture accurately the large NO$_2$ gradients at low altitudes, particularly for polluted regions where the measurement sensitivity of the satellite decreases significantly towards the surface.

Figure 9 shows the absolute and relative differences in tropospheric NO$_2$ columns retrieved by altering the model resolutions for IFS(CBA) a priori NO$_2$ profiles in February 2010. In Fig. 9a, the increase of the spatial resolution (1° vs. 0.7° grid) changes the tropospheric NO$_2$ columns by up to $7 \times 10^{14}$ molec/cm$^2$ or 20% for polluted regions, in agreement with previous case studies or regional retrievals (Heckel et al., 2011; Lin et al., 2014; Kuhlmann et al., 2015). Larger relative differences are observed over cities surrounded by rural areas, coastal regions, isolated islands, and shipping lanes, where the use of high

spatial resolutions captures more accurately the NO$_2$ emission and chemistry for a priori profiles. In Fig. 9b, the improvement in the vertical resolution (34 vs. 69 layers) enhances the tropospheric NO$_2$ columns by up to $5 \times 10^{14}$ molec/cm$^2$ or 15%. Increasing the number of layers generally better resolves the NO$_2$ vertical variation, especially for the lowest model layers



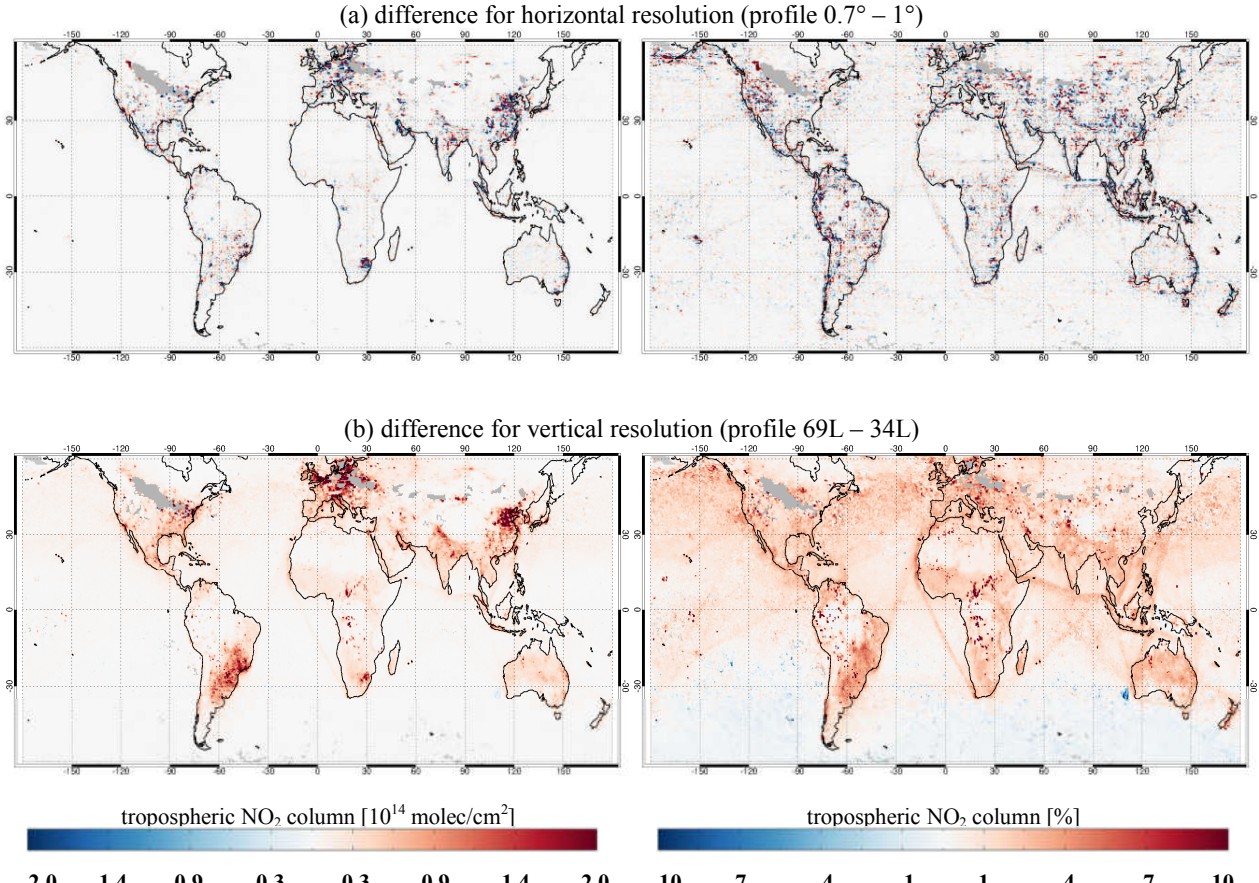

**Figure 9.** Absolute and relative differences in tropospheric NO$_2$ columns retrieved by altering the model resolutions for IFS(CBA) a priori NO$_2$ profiles in February 2010. The tropospheric NO$_2$ columns are compared for 1° and 0.7° grid (a) and for 34 and 69 layers (b). Only measurements with cloud radiance fraction < 0.5 are included.

where the box-AMF decreases significantly in the polluted cases. Consequently, the tropospheric AMFs are lower and the tropospheric NO$_2$ columns are higher for polluted regions. For unpolluted regions, the differences are generally small (within $\pm 2 \times 10^{14}$ molec/cm$^2$ or $\pm 3\%$). In addition, the use of different temporal resolution (2-hours vs. 1-hour time step) generally has a negligible impact on the tropospheric NO$_2$ columns (less than $2 \times 10^{14}$ molec/cm$^2$ or 3%, not shown).

### 3.3 Cloud correction

For cloudy scenarios, the retrieval of tropospheric NO$_2$ is affected by the cloud parameters due to the variation of scene albedo and the photon path redistribution in the troposphere. As described in Sect. 2, the cloudy-sky AMFs are calculated with the independent pixel approximation using GOME-2 cloud parameters: radiometric cloud fraction from OCRA and cloud top pressure (cloud top height) and cloud albedo (cloud optical depth) from ROCINN. To improve the cloud correction in





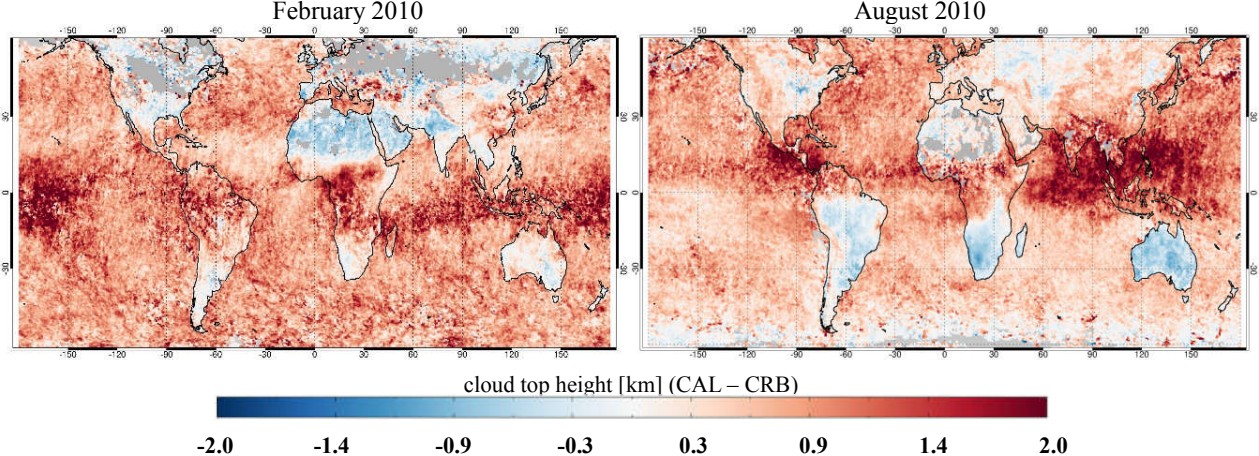

February 2010 — August 2010

cloud top height [km] (CAL − CRB)

-2.0    -1.4    -0.9    -0.3    0.3    0.9    1.4    2.0

**Figure 10.** Differences in cloud top heights retrieved using ROCINN_CAL and ROCINN_CRB cloud model in February and August 2010. Only measurements with cloud fraction < 0.3 are included. Observations with fitting RMS > $1 \times 10^{-4}$ or number of iterations > 20 are filtered out.

our $NO_2$ retrieval, the CAL model from ROCINN cloud algorithm (Loyola et al., 2018), with the clouds treated as optically uniform layers of light-scattering water droplets, is applied. The CAL model is more representative of the real situation than the CRB model (with the clouds idealized as Lambertian reflectors with zero transmittance) by allowing the penetration of

photons through the cloud layer. This treatment takes the multiple scattering of light inside the cloud and the contribution of the atmospheric layer between the cloud bottom and the ground into account.

Figure 10 shows the differences in cloud top heights obtained with the CRB and CAL model from GOME-2 measurements in February and August 2010. Consistent with Loyola et al. (2018) (see Fig 3 and 13 therein), the cloud top heights from CAL are generally higher by on average ∼0.9 km. Stronger increases (up to 2 km) are found over regions with thick and high clouds,

such as the Intertropical Convergence Zone and the Western Hemisphere Warm Pool, very similar to Lelli et al. (2012) (see Fig. 12 therein). In general, the CRB-based cloud retrieval underestimates the cloud top height due to the neglect of oxygen absorption throughout a cloud layer (Vasilkov et al., 2008) and thus the misinterpretation of the smaller top-of-atmosphere reflectance as a lower cloud layer (Saiedy et al., 1967). Additionally, since the enhanced multiple scattering is not fully taken into account in the CRB-based cloud retrieval, the retrieved cloud height is normally close to the middle, i.e., the optical centroid of clouds (Ferlay et al., 2010; Richter et al., 2015).

In Fig. 10, higher cloud top heights are found using CRB mainly over land surfaces characterised by the presence of a large amount of absorbing aerosols, for instance, over regions with strong desert dust emissions, such as the Sahara, the Arabian Desert, and the deserts in Australia, as well as regions with strong biomass burning emissions, such as South America, South

Africa, and Southeast Asia. Over these areas, ROCINN likely retrieves an effective aerosol height close to the top of aerosol layer, depending on the type of absorbing aerosols and on aerosol optical depth. The presence of strongly absorbing aerosols, which typically have large aerosol optical depth and/or locate at high altitudes (up to ∼8 km), reduces the fraction of photons





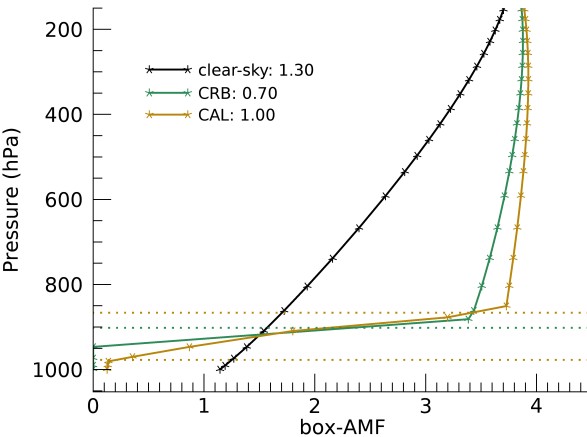

**Figure 11.** The box-AMFs for clear and cloudy sky using ROCINN_CAL and ROCINN_CRB cloud model over Italy (45.3° N, 11.2° E) on 1 February 2010. The tropospheric AMF is given next to each label. The ROCINN_CRB cloud top pressure is shown as a horizontal green line, and the ROCINN_CAL cloud top and base pressure are shown as horizontal brown lines. Cloud radiance fraction = 0.47, cloud optical depth = 6.85, SZA = 69°, VZA = 3°, RAA = 42°.

reaching the lowest part of the atmosphere. In order to approximate this shortened light path, the CRB-based cloud retrieval has to put the Lambertian reflector at a higher altitude (Wang et al., 2012; Chimot et al., 2016). This effect is larger for aerosol layers at higher altitudes and dependent also on geometry parameters like SZA, on surface properties like surface albedo, and on the accuracy of radiometric cloud fractions from OCRA.

To apply the CAL cloud model in our $NO_2$ AMF calculation, a single scattering albedo of 1 and an asymmetry parameter of 0.85 for water clouds are assumed for the radiative transfer calculation, consistent with the values used in the cloud retrieval (Loyola et al., 2018). Cloud observations with fitting root mean square (RMS) $> 1 \times 10^{-4}$ or number of iterations $> 20$ are filtered out. The $NO_2$ box-AMFs are derived through the pixel-specific radiative transfer calculation instead of the interpolation from a LUT with fixed reference points, which requires no projection from the layer coordinate of $NO_2$ profile to the coordinate assumed in the LUT and requires no linear interpolation based on the model parameters.

Figure 11 shows an example of the derived box-AMFs for clear and cloudy sky using the CRB and CAL model over Italy (45.3° N, 11.2° E) on 1 February 2010. The cloud information and the calculated tropospheric AMFs are also reported. Compared to the clear-sky box-AMFs, the CAL-based cloudy-sky box-AMFs increase above the cloud layer (albedo effect) and decrease below the cloud layer (shielding effect). Compared to the CRB model, the use of CAL model takes account of the sensitivities inside and below the cloud layer and increases the cloudy-sky AMF by 0.3, which consequently decreases the retrieved tropospheric $NO_2$ column by $2.5 \times 10^{15}$ molec/cm$^2$ (12%), based on the polluted $NO_2$ profile with most of the $NO_2$ concentration located near the surface.

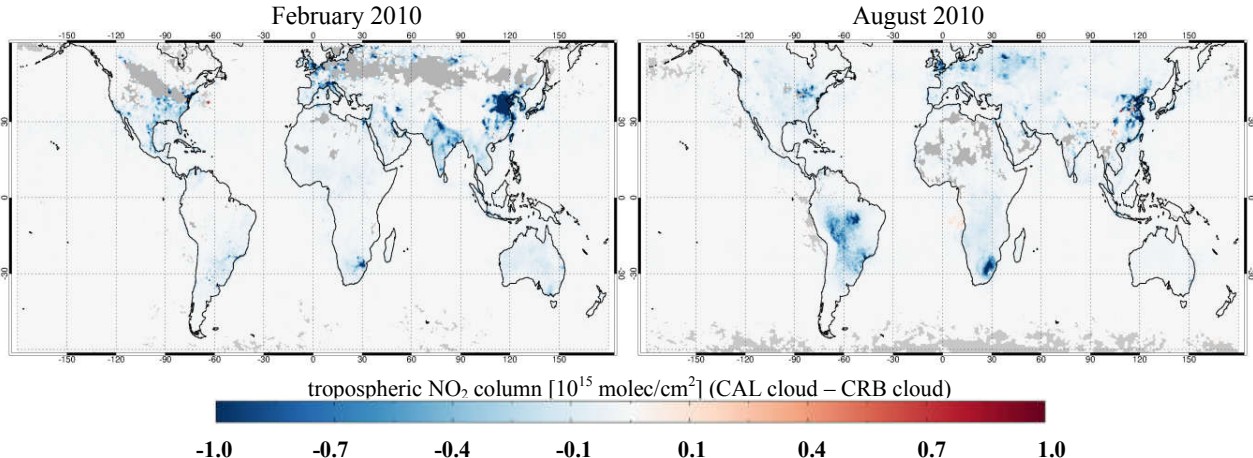

**Figure 12.** Differences in GOME-2 tropospheric $NO_2$ columns retrieved using ROCINN_CRB and ROCINN_CAL cloud model in February and August 2010. Only measurements with cloud radiance fraction < 0.5 are included. Cloud observations with fitting RMS > $1 \times 10^{-4}$ or number of iterations > 20 are filtered out.

Figure 12 shows the differences in tropospheric $NO_2$ columns retrieved using CRB and CAL model in February and August 2010. The use of CAL model decreases the tropospheric $NO_2$ columns by more than $1 \times 10^{14}$ molec/cm$^2$ over polluted regions. Larger values are found in winter (up to $3 \times 10^{15}$ molec/cm$^2$), when most of the $NO_2$ concentrations are located at the surface and the cloud fractions are generally larger due to the seasonal variation of clouds.

### 3.4 Combined impact

Figure 13 shows the tropospheric $NO_2$ columns retrieved using the improved AMF calculation and the differences with the reference data in February and August 2010. Larger differences are found in winter over the polluted regions. For instance, the tropospheric $NO_2$ columns are reduced by more than $1 \times 10^{15}$ molec/cm$^2$ over China and India in February and Brazil and South Africa in August. Increased values are found e.g. over Mexico, Argentina, and Russia.

Table 3 summarizes the individual changes and combined effect of our improved AMF calculation on the retrieved tropospheric $NO_2$ columns over western Europe, eastern China, eastern US, and central Africa. Increases in GOME-2 surface albedo reduce the tropospheric $NO_2$ columns by 2-6%. The use of IFS(CBA) a priori $NO_2$ profiles affects (mostly increases) the tropospheric $NO_2$ columns by up to 21%, and the use of CAL cloud parameters decreases the values by up to 14%. The combined effect of individual improvements yields to a change of tropospheric $NO_2$ columns on average within ±15% in winter and ±5% in summer over polluted regions.

The uncertainty in the improved AMF calculation is likely reduced comparing to the reference retrieval, considering the improved surface albedo dataset, a priori $NO_2$ profiles, and cloud parameters, which are the main causes of AMF uncertainty (Lorente et al., 2017). The uncertainty in the AMF calculation for polluted conditions is estimated to improve from 10-45% for the reference retrieval to the 10-35% range for this work.


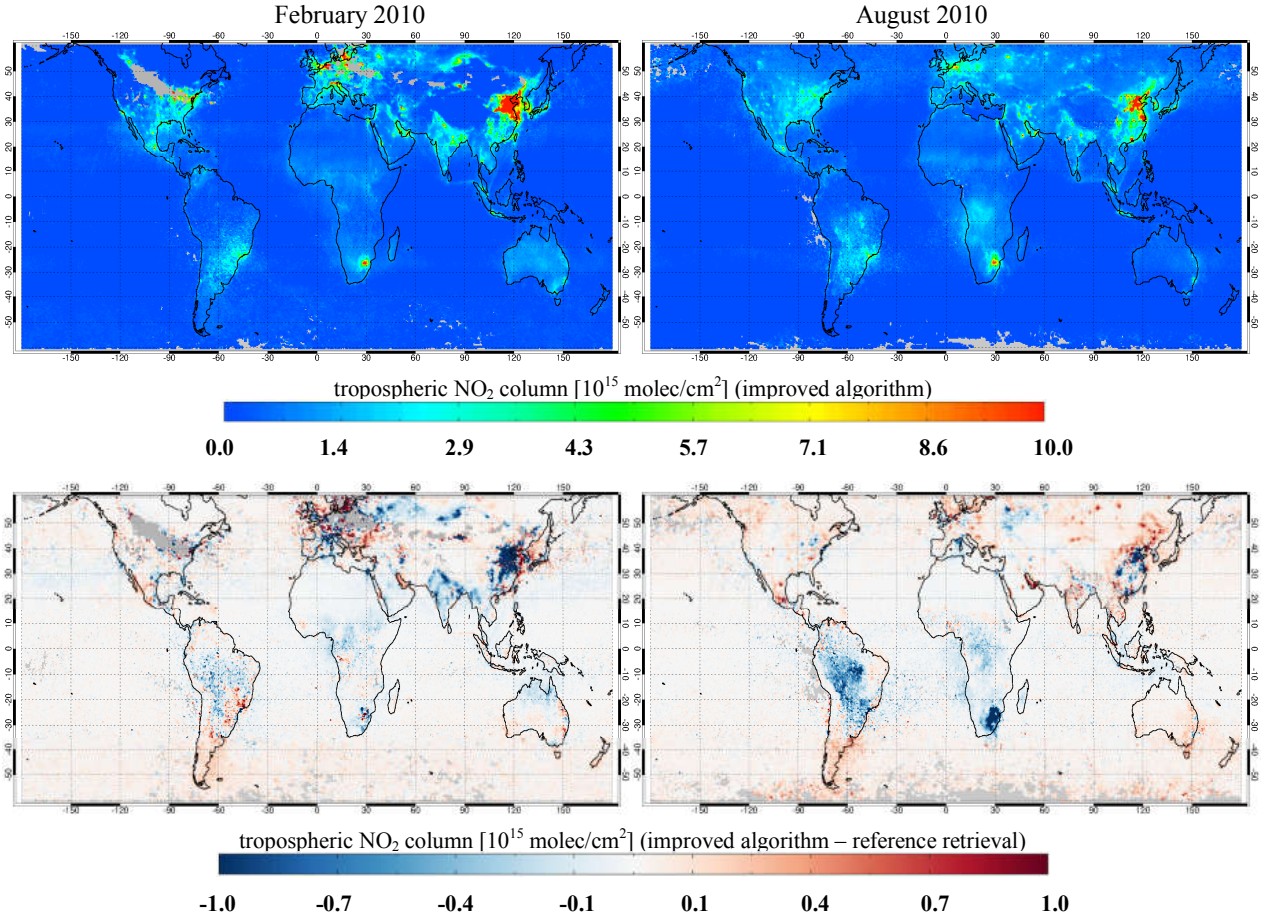

**Figure 13.** GOME-2 tropospheric $NO_2$ columns retrieved using the improved algorithm and the differences with the reference data in February and August 2010. Only measurements with cloud radiance fraction < 0.5 are included. Cloud observations with fitting RMS > $1 \times 10^{-4}$ or number of iterations > 20 are filtered out.

## 4 Implicit aerosol correction

Aerosols can increase or decrease the sensitivity to tropospheric $NO_2$, depending on the $NO_2$ and aerosol vertical distribution, and the optical and physical properties of the particles (Martin et al., 2003; Leitão et al., 2010). Since the OCRA/ROCINN cloud retrieval does not distinguish between clouds and aerosols, the aerosol effect is assumed to be corrected implicitly in the AMF calculation via the effective cloud parameters (i.e., aerosols are treated as clouds).

Figure 14 and 15 show the land surface RGB image with active fire locations from the Moderate Resolution Imaging Spectroradiometer (MODIS) on board the Terra (10:30 LT) and OCRA/ROCINN cloud products measured by GOME-2 (9:30 LT) over eastern China and central Africa on a given day, respectively. The MODIS dataset (https://worldview.earthdata.nasa. gov/) describes the cloud or aerosol amount and fire locations (for central Africa). For both regions, a large amount of aerosol



**Table 3.** Individual changes and combined effect in tropospheric $NO_2$ columns relative to reference retrievals for western Europe (44° N-53° N, 0° E-7° E), eastern China (21° N-41° N, 110° E-122° E), eastern US (30° N-45° N, 70° W-90° W), and central Africa (5° S-15° S, 10° E-30° E).

|  | surface albedo | a priori $NO_2$ profile | cloud correction | combined effect |
|---|---|---|---|---|
| Europe (February 2010) | -2.2% | +19.6% | -9.3% | +2.0% |
| China | -5.9% | +0.7% | -12.1% | -13.3% |
| US | -4.6% | +15.6% | -12.2% | +8.9% |
| Africa | -2.1% | -1.2% | -3.3% | -5.8% |
| Europe (August 2010) | -3.6% | +20.5% | -9.4% | +1.1% |
| China | -5.6% | +15.9% | -14.0% | -2.3% |
| US | -4.3% | +10.1% | -9.7% | +1.1% |
| Africa | -3.8% | -0.6% | -5.4% | -4.8% |

loads are found in the RGB image for cloud-free areas, e.g. Beijing-Tianjin-Hebei economic region in eastern China and burning regions across central Africa, where the aerosol loads are identified as thin clouds (cloud optical depth of ∼5) near the surface (cloud top height of ∼3 km) with cloud fractions up to 0.18.

Therefore, we assume that the thin clouds near the surface from the OCRA/ROCINN cloud products are attributed to aerosol loads for measurements with cloud radiance fraction < 0.5 or cloud fraction < 0.2, and we evaluate the accuracy of implicit aerosol correction by comparing it with explicit aerosol correction. For that purpose, the explicit correction for aerosols is implemented using ground-based aerosol observations in Xianghe (39.75° N, 116.96° E), which is a suburban site surrounded by heavily industrialized areas in northeastern China (Clémer et al., 2010; Hendrick et al., 2014; Vlemmix et al., 2015), and in Bujumbura (3.38° S, 29.3° E), which is located in the Central African country of Burundi with intensive biomass burning activities in the surroundings (Gielen et al., 2017), as indicated in Fig. 14 and 15, respectively. Our analysis is further limited to satellite measurements with cloud optical depth < 5 and cloud top height < 3 km to reduce the cloud contamination. With this selection, the aerosol concentrations are generally low or moderate (aerosol optical depth < 1).

The explicit modelling of aerosol scattering and absorption for the AMF calculation is implemented by introducing the aerosol optical properties (i.e., single scattering albedo and phase function) and vertical distributions (i.e., extinction vertical profiles) in the radiative transfer calculation. The single scattering albedo describes the fraction of the aerosol light scattering over the extinction, and the phase function describes the angular distribution of scattered light intensity. In this study, we apply the Henyey-Greenstein phase function with an asymmetry parameter (the first moment of phase function) describing the asymmetry between forward and backward scattering. A long-term statistics of single scattering albedo and asymmetry parameter at 440 nm is derived for Xianghe and Bujumbura using the version 3 level 2.0 inversion products from the sun photometer radiance measurements at AERONET (Holben et al., 1998; Giles et al., 2019) (http://aeronet.gsfc.nasa.gov/). Monthly mean



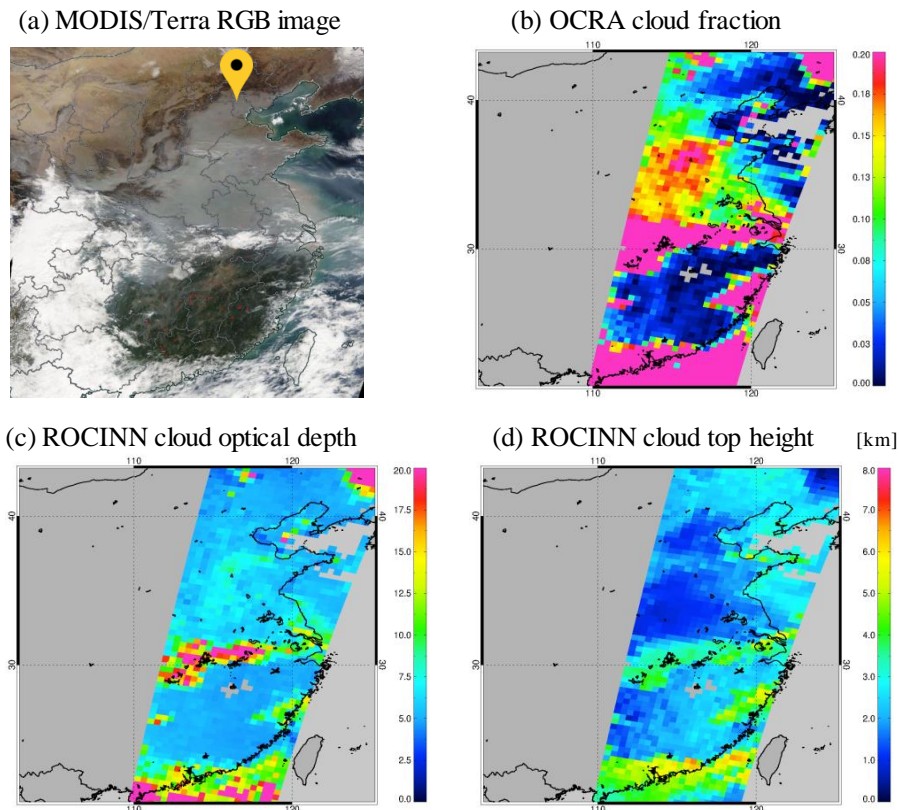

**Figure 14.** MODIS/Terra RGB image (a), GOME-2 OCRA cloud fraction (b), GOME-2 ROCINN_CAL cloud optical depth (c), and GOME-2 ROCINN_CAL cloud top height (d) over eastern China on 21 November 2013. Cloud observations with fitting RMS $> 1 \times 10^{-4}$ or number of iterations $> 20$ are filtered out. The yellow location symbol in MODIS image indicates the Xianghe station ($39.75°$ N, $116.96°$ E), and the red dots indicate fires.

parameters are calculated based on up to seven years of observations (2010-2016 for Xianghe and 2013-2016 for Bujumbura) available within $\pm 1$ h of the GOME-2 overpass time (9:30 LT) for each month.

Xianghe is located $\sim$60 km south-east of Beijing, belonging to the highly urbanized Beijing-Tianjin-Hebei economic region on the North China Plain with heavy anthropogenic aerosol emissions, especially in winter due to the enhanced domestic heating. Mixtures of desert dust with the urban-industrial aerosols affect the regions mainly in spring (March-May). Based on the monthly climatology of AERONET measurements, the single scattering albedo in Xianghe is on average 0.91 with a maximum in July (0.96) and low values in winter ($\sim$0.87), which are mostly related to the black carbon emissions (Yang et al., 2011). The asymmetry parameter ranges between 0.7 and 0.75, consistent with the values from the urban aerosol models in East Asia (Lee and Kim, 2010).



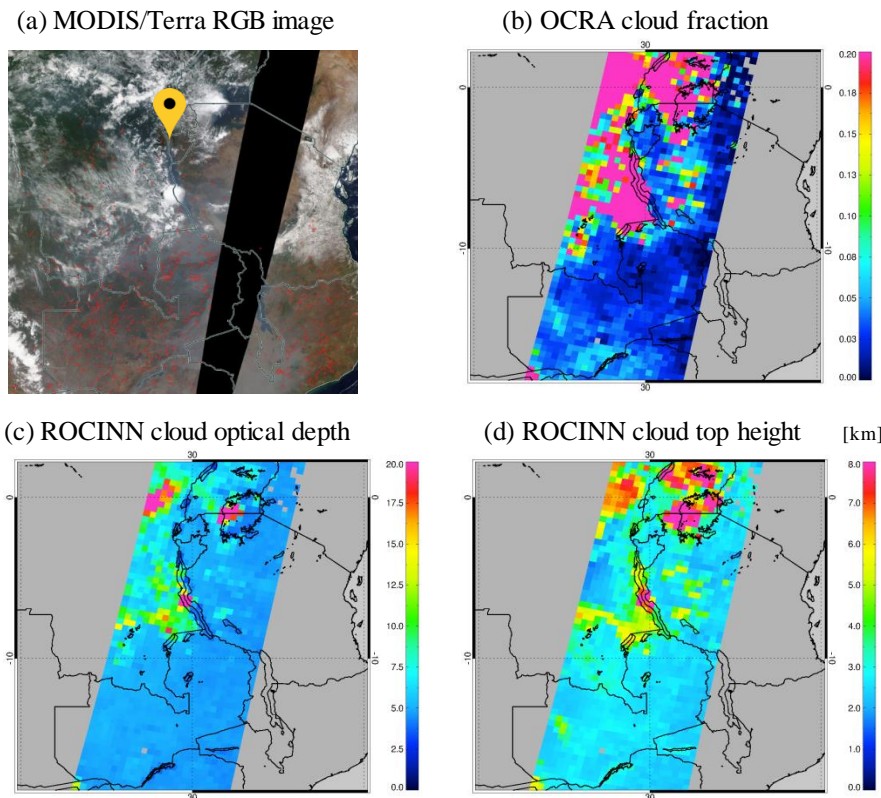

**Figure 15.** Similar as Fig. 14 but for central Africa on 9 September 2015. The yellow location symbol in MODIS image indicates the Bujumbura station (3.38° S, 29.3° E).

Bujumbura is located in tropical Africa that is typically affected by the biomass burning emissions, mainly during the local dry seasons (June-August and January-February), and to a lesser extend, by the anthropogenic emissions, throughout the year with negligible seasonal variations. The single scattering albedo in Bujumbura is higher in March-May (∼0.9), related to the major rain season, and lower in July-August and December-January (0.83-0.87), coinciding with the intensive agricultural activities and transport of forest fire emissions in the surrounding regions (Gielen et al., 2017). The asymmetry parameter is on average 0.69, in agreement with values in biomass burning aerosol models (Torres et al., 2013).

The collocated aerosol extinction vertical profiles at 477 nm are taken from the MAXDOAS measurements in Xianghe from March 2010 to December 2016 (Clémer et al., 2010) and in Bujumbura from December 2013 to December 2015 (Gielen et al., 2017). The MAXDOAS data is used to derive aerosol information based on the oxygen collision complexes ($O_4$) absorption, since the $O_4$ vertical profile is generally constant and thus capable of describing the influence of aerosol scattering and absorption on photon path (Wagner et al., 2004; Frieß et al., 2006). The MAXDOAS technique can reliably determine the aerosol extinction profiles in the lower troposphere (Frieß et al., 2016), where most aerosols are located over Xianghe and Bujum-





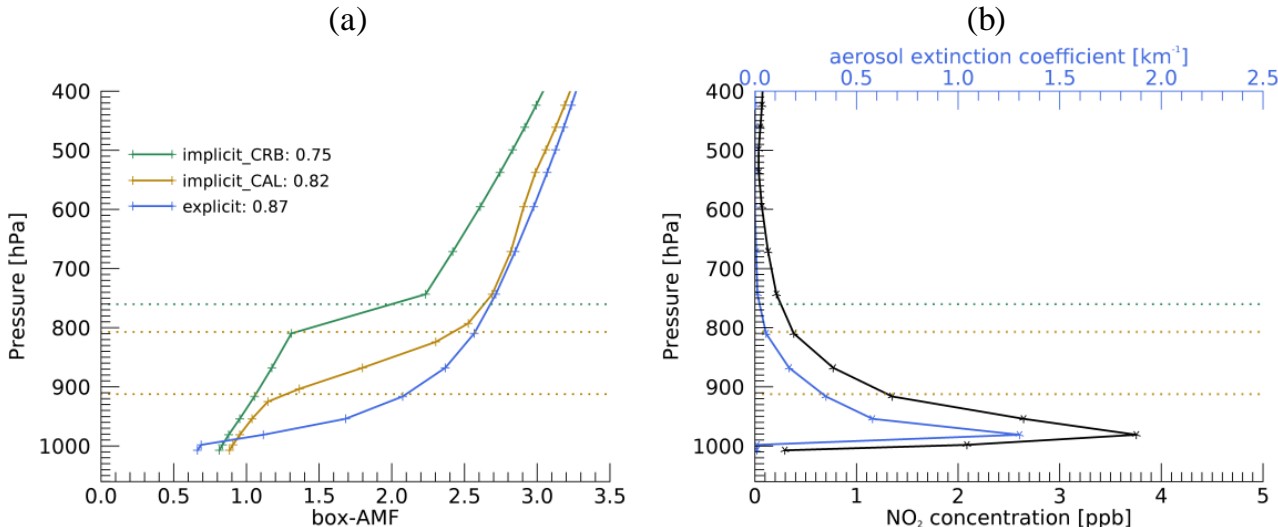

**Figure 16.** The $NO_2$ box-AMFs from the explicit aerosol correction and implicit correction using ROCINN_CRB and ROCINN_CAL cloud model (a) and the TM5-MP $NO_2$ profiles and the MAXDOAS aerosol extinction profiles used in the retrievals (b) at the Xianghe station on 21 November 2013. The tropospheric AMF is given next to each label. The ROCINN_CRB cloud top pressure is shown as a horizontal green line, and the ROCINN_CAL cloud top and base pressure are shown as horizontal brown lines. Cloud radiance fraction = 0.28, cloud optical depth = 4.96, aerosol optical depth = 0.66, SZA = 66°, VZA = 17°, RAA = 133°.

bura. We collocate the space- and ground-based measurements by selecting GOME-2 pixels within 50 km of the stations and averaging the MAXDOAS aerosol profiles within ±1 h of the GOME-2 overpass time (9:30 LT).

Figure 16 shows typical $NO_2$ box-AMFs, simulated TM5-MP $NO_2$ profile, and MAXDOAS aerosol extinction profile for

Xianghe on 21 November 2013. The MAXDOAS aerosol profile follows an exponentially decreasing shape with a peak of aerosol loads close to the ground (950 hPa or 0.4 km). The $NO_2$ follows the same profile shape and is well mixed with aerosol. Depending on seasonal variation, local emission, and transport process, aerosol profiles with peak at elevated heights (up to 900 hPa or 1 km) are also observed (not shown) due to the long residence time. The discontinuity of box-AMFs corrected using the CRB cloud model is introduced by the effective clouds (see Eq. (2)), below which the cloudy box-AMFs are zero.

Due to the overestimated cloud altitudes from the CRB-based cloud retrieval (see Sect. 3.3), the CRB-based implicit aerosol correction underestimates the tropospheric AMF by 14%, and this bias is largely reduced by applying the CAL cloud model (6%), which brings a gradual reduction in box-AMFs towards the surface, agreeing reasonably better with the shape from explicit aerosol correction. Figure 17 shows the same data as Fig. 16 but for Bujumbura on 9 September 2015. Compared to the data in Xianghe, the aerosol profile in Bujumbura shows a smaller amount but an uplifted layer of aerosol loads (820 hPa or

1.8 km), while $NO_2$ continues to peak at the surface. The difference in AMF between implicit and explicit aerosol correction decreases from 15% using CRB to 5% using CAL.





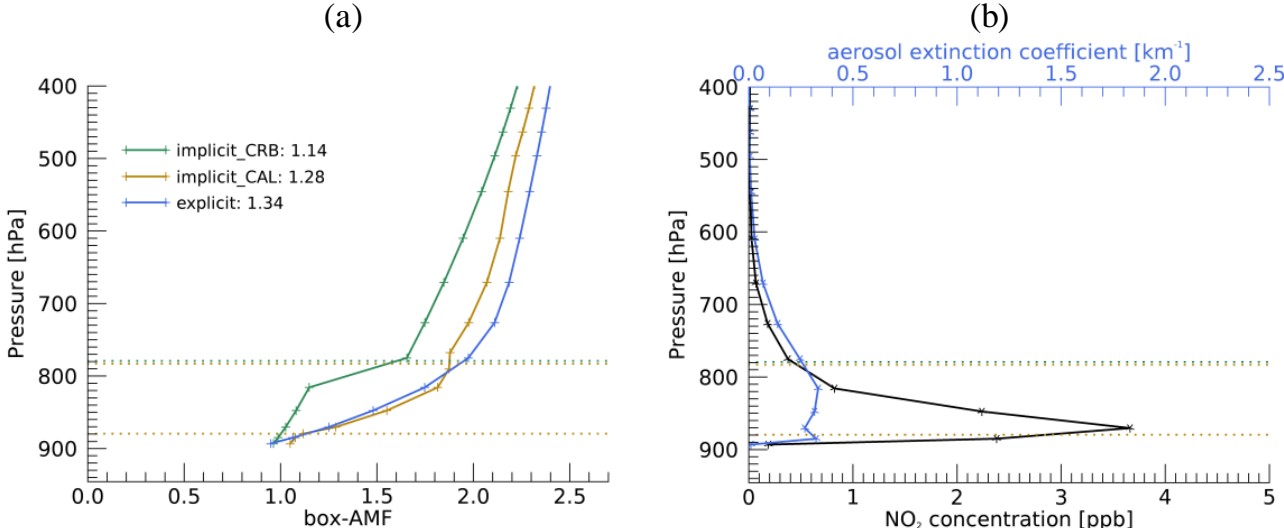

**Figure 17.** Similar as Fig. 16 but for the Bujumbura station on 9 September 2015. Cloud radiance fraction = 0.18, cloud optical depth = 4.83, aerosol optical depth = 0.58, SZA = 40°, VZA = 14°, RAA = 155°.

Figure 18 presents the relative biases in tropospheric $NO_2$ columns retrieved assuming no aerosol correction and assuming implicit aerosol correction via the CRB and CAL cloud model for Xianghe from March 2010 to December 2016. Only measurements with cloud radiance fraction < 0.5, cloud optical depth < 5, and cloud top height < 3 km are included. The relative

biases introduced by assuming no aerosol correction vary between -30% to 31% with an average of 7% for GOME-2 pixels, in agreement with previous studies focusing on the industrialized part of eastern China (Ma et al., 2013; Lin et al., 2014, 2015; Kuhlmann et al., 2015; Chimot et al., 2016; Wang et al., 2017). Resulted from the overestimated shielding effect, the tropospheric $NO_2$ columns retrieved using CRB-based implicit aerosol correction are on average 33% larger than using explicit aerosol correction, and the differences are largely reduced by applying the CAL cloud model (9%). Enhanced differences are

found for larger cloud radiance fraction, probably due to the increased pollution level ($NO_2$ columns) comparing to the clear sky (Richter et al., 2017), as the cloud (radiance) fraction is highly correlated with the MAXDOAS aerosol optical depth (correlation coefficient of 0.7 and regression slope of 0.17, not shown). Figure 19 shows the same data but for Bujumbura from December 2013 to December 2015. The explicit aerosol correction yields tropospheric $NO_2$ columns on average 6% smaller than the clear-sky tropospheric $NO_2$ columns, consistent with Martin et al. (2003); Castellanos et al. (2015). The average dif-

ference between the tropospheric $NO_2$ columns from the implicit and explicit aerosol correction decreases from 15% using CRB model to 5% using CAL model.

In Fig. 18 and 19, the relative biases introduced by the CAL-based implicit aerosol correction are close to the values assuming no aerosol correction, addressing the complexities related to the tropospheric $NO_2$ measurements in the presence of aerosols. In combination with the cloud model error, errors related to the implicit aerosol correlation can result from the different radiative





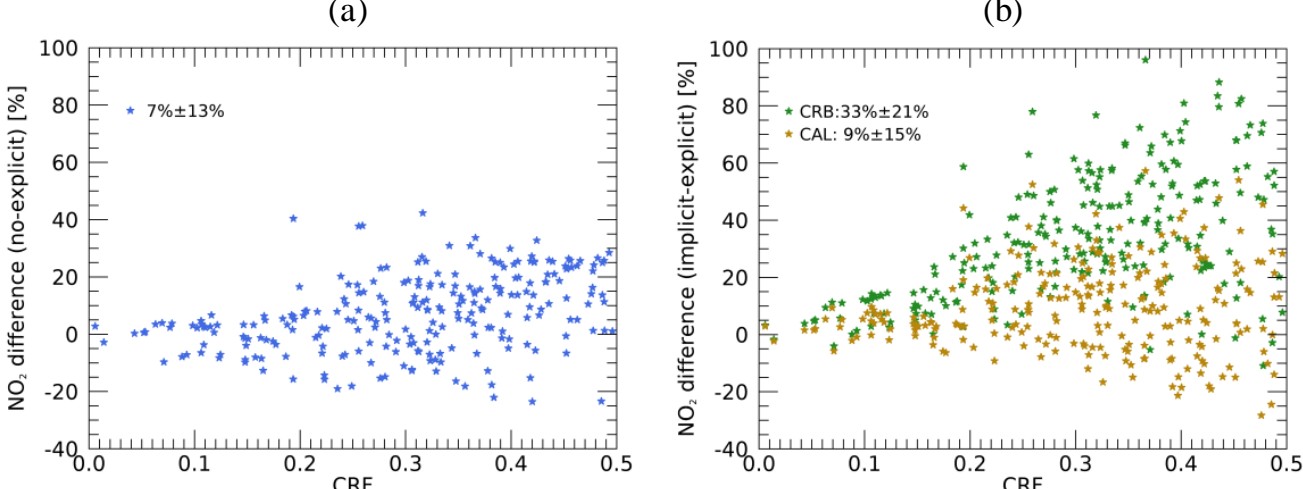

**Figure 18.** Scatter plot for relative biases in GOME-2 tropospheric $NO_2$ columns assuming no aerosol correction (a) and assuming implicit aerosol correction through ROCINN_CRB and ROCINN_CAL cloud model (b) with respect to the cloud radiance fraction at the Xianghe station from March 2010 to December 2016. Only measurements with cloud radiance fraction $< 0.5$, cloud optical depth $< 5$, and cloud top height $< 3$ km are included. Cloud observations with fitting RMS $> 1 \times 10^{-4}$ or number of iterations $> 20$ are filtered out. The mean value and standard deviation are given next to each label.

effect of scattering clouds and absorbing aerosols and the different characteristic sizes and phase functions of clouds and aerosols in general. The errors may be additionally enhanced in the presence of actual clouds. Therefore, future works include the further quantitative interpretation of OCRA/ROCINN cloud parameters for aerosol-dominated scenes and the impact on $NO_2$ retrieval algorithm.

## 5   Tropospheric $NO_2$ validation

A validation of our improved GOME-2 tropospheric $NO_2$ columns is performed with BIRA-IASB ground-based MAXDOAS measurements at the Xianghe station from March 2010 to December 2016. For the validation of GOME-2 measurements, the satellite data is filtered for clouds (cloud radiance fraction $< 0.5$), and the closest valid pixel within 50 km of the stations is compared to the ground-based MAXDOAS data, which is linearly interpolated to the GOME-2 overpass time (9:30 LT), if original data exists within $\pm 1$ hour. As introduced in Sect. 4, Xianghe is a typical suburban station adequate for GOME-2

tropospheric $NO_2$ validation (Liu et al., 2019b). Urban stations are generally underestimated by GOME-2 data due to the averaging of a local source over a pixel size (Pinardi et al., 2015; Pinardi, in preparation).

     Figure 20 shows the time series and scatter plot of the daily and monthly means comparison between the improved GOME-2 tropospheric $NO_2$ columns and the ground-based MAXDOAS measurements in Xianghe, including the statistical information





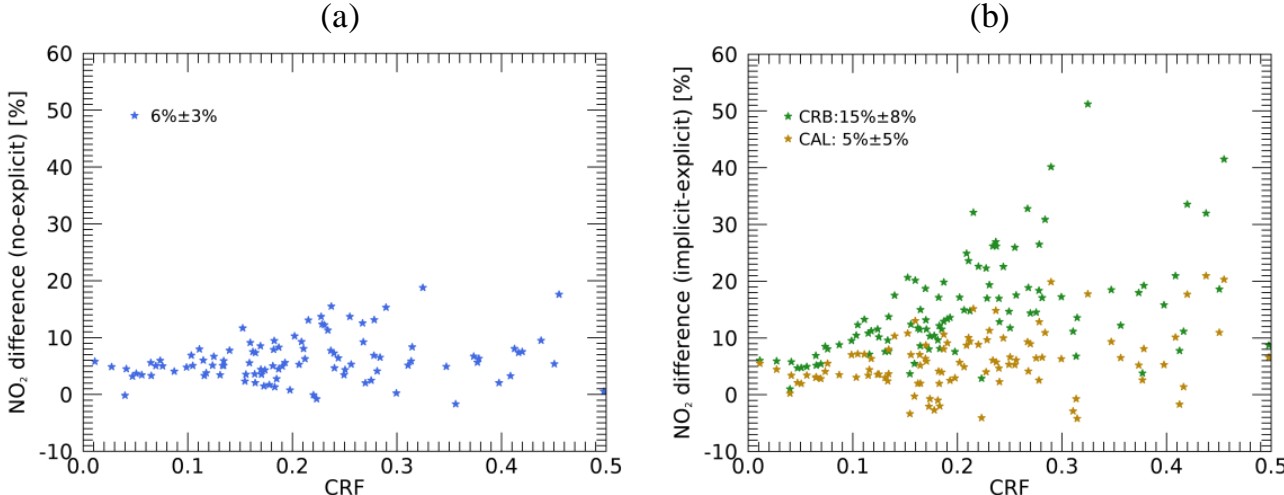

**Figure 19.** Similar as Fig. 18 but for the Bujumbura station from December 2013 to December 2015.

on the correlation coefficient, slope, and intercept of orthogonal regression analysis. The monthly mean values from the GOME-

2 and MAXDOAS measurements indicate good agreement with similar seasonal variations in tropospheric $NO_2$ column. A correlation coefficient of 0.94, a regression slope of 0.69 ($\pm$0.03) and an intercept of 0.41 ($\pm$0.06)$\times 10^{15}$ molec/cm$^2$ are derived when comparing the monthly mean values. These results are qualitatively similar to previous validation exercises at other sites, other satellites, and other $NO_2$ products (Celarier et al., 2008; Kramer et al., 2008; Chen et al., 2009; Irie et al., 2012; Ma et al., 2013; Wu et al., 2013; Kanaya et al., 2014; Wang et al., 2017; Drosoglou et al., 2017, 2018). Similar figures for previous GDP products can be found in Liu et al. (2019b) and AC-SAF validation website (http://cdop.aeronomie.be/validation/valid-results).

Figure 21 presents the daily and monthly mean absolute and relative differences of GOME-2 and MAXDOAS measurements. The differences are on average within $\pm 1 \times 10^{16}$ molec/cm$^2$ with a mean difference of $-3.7 \times 10^{15}$ molec/cm$^2$. The $NO_2$ levels

5   are underestimated by 9.9% by GOME-2 with a standard deviation of $\pm$21%, mostly explained by the relatively low sensitivity of space-borne measurements near the surface, the gradient-smoothing effect, and the aerosol shielding effect. These effects are often inherent to the different measurements types or the specific conditions of the validation sites and also to the remaining impact of structural uncertainties (Boersma et al., 2016), such as the impact of the choices of the a priori $NO_2$ profiles and/or the albedo database assumed for the satellite AMF calculations.

10   To summarise the improvements of each of the changes discussed in previous sections, Table 4 reports the statistical results including the biases and regression analysis for the use of different surface albedo and a priori $NO_2$ profiles at the Xianghe station for completely clear sky (cloud radiance fraction = 0). Compared to the reference retrieval (based on GOME-2 surface LER climatology and TM5-MP a priori profile), better results are obtained with the improved algorithm (based on surface DLER dataset and IFS(CBA) a priori profile) with a median difference of $-1.0 \times 10^{15}$ molec/cm$^2$, which will be used to

15   further test for aerosol correction type below.

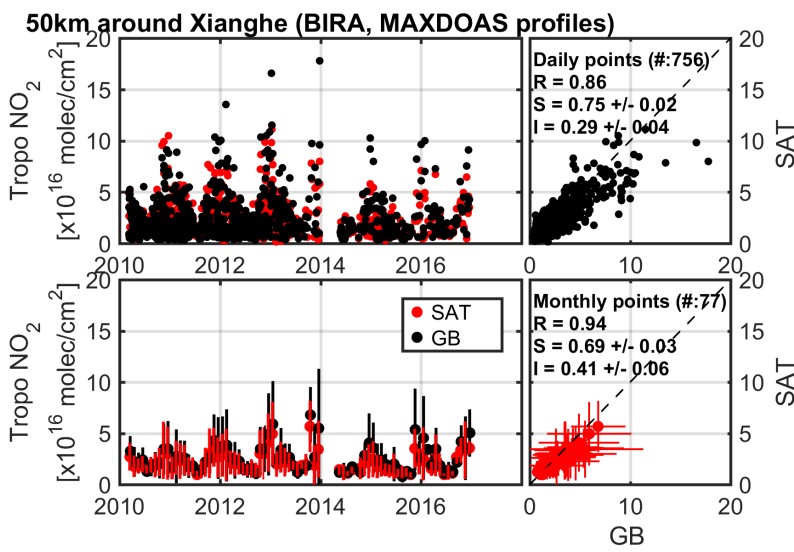

**Figure 20.** Daily and monthly mean time series and scatter plot of GOME-2 and MAXDOAS tropospheric NO$_2$ columns (mean value of all the pixels within 50km around Xianghe).

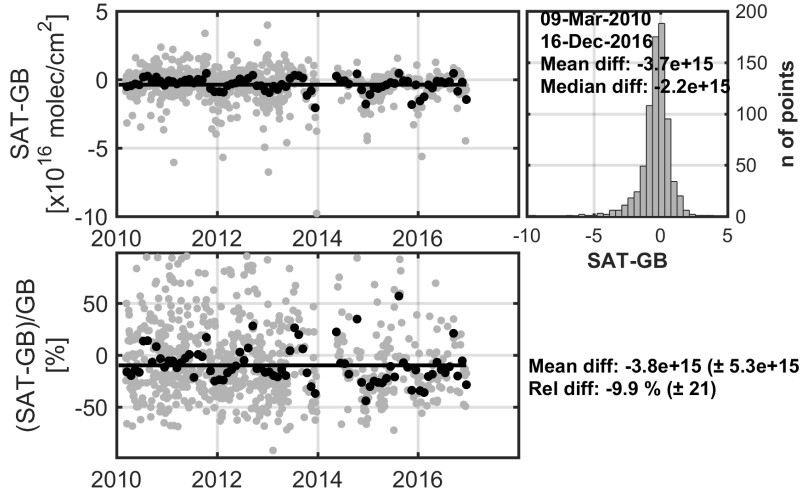

**Figure 21.** Daily and monthly mean absolute and relative GOME-2 and MAXDOAS time series differences for the Xianghe station. The histogram of the daily differences is also given, with the mean and median difference. The total time series absolute and relative monthly differences are given outside the panels.



**Table 4.** Mean difference (MD), median difference (AD) (SAT-GB in $10^{15}$ molec/cm$^2$), standard deviation (STDEV), and correlation coefficient R and regression parameters (slope S and intercept I) of the orthogonal regression for the daily GOME-2 tropospheric NO$_2$ product when comparing to MAXDOAS data at Xianghe. Intermediate results for different surface albedo and a priori NO$_2$ profiles are reported for completely clear sky (cloud radiance fraction = 0) for a total of 73 GOME-2 pixels.

| surface albedo | LER | DLER | DLER |
|---|---|---|---|
| a priori NO$_2$ profile | TM5-MP | TM5-MP | IFS(CBA) |
| MD±STDEV ($\times 10^{15}$) | -2.5±7.2 | -2.6±7.1 | -1.9±7.2 |
| AD ($\times 10^{15}$) | -1.9 | -1.7 | -1.0 |
| R | 0.63 | 0.64 | 0.63 |
| S | 0.60±0.09 | 0.60±0.08 | 0.63±0.09 |
| I | 0.30±0.12 | 0.29±0.12 | 0.31±0.12 |

**Table 5.** Similar as Table 4 but for different aerosol corrections for aerosol-dominated conditions (cloud radiance fraction < 0.5, cloud optical depth < 5, and cloud top height < 3 km) for a total of 146 GOME-2 pixels. Results are calculated using DLER surface albedo and IFS(CBA) a priori profiles.

| aerosol correction | no | implicit_CAL |
|---|---|---|
| MD±STDEV ($\times 10^{15}$) | -4.0±10.0 | -2.7±9.4 |
| AD ($\times 10^{15}$) | -2.8 | -2.3 |
| R | 0.83 | 0.86 |
| S | 0.72±0.04 | 0.91±0.05 |
| I | 0.34±0.09 | -0.05±0.10 |

Table 5 presents the statistical results for the retrievals with no aerosol correction and CAL-based implicit aerosol correction at the Xianghe station for aerosol-dominated cases (cloud radiance fraction < 0.5, cloud optical depth < 5, and cloud top height < 3 km). Consistent with Fig. 18, the GOME-2 NO$_2$ columns retrieved using the CAL-based implicit aerosol correction are higher than the results assuming no aerosol correction, which improve the biases relative to the MAXDOAS measurements, as well as the standard deviation, correlation coefficient, and regression parameters. We note here that all the validation results in this work show a significant improvement compared to the current operational GDP 4.8 product (Pinardi et al., 2015; Pinardi, in preparation; Liu et al., 2019b) in the AC-SAF context (Hassinen et al., 2016).

# 6   Conclusion

The operational GOME-2 NO$_2$ dataset, generated with the GDP algorithm at DLR, has been introduced in detail by Valks
et al. (2011, 2017) and successfully applied in many studies. An improved AMF calculation with more accurate knowledge





of surface albedo, a priori NO$_2$ profile, as well as cloud and aerosol correction is described in this paper and expected to be implemented in an upcoming version of GDP in combination with Liu et al. (2019b).

The viewing angle-dependency of surface albedo is taken into account by improving the currently used GOME-2 surface LER climatology (Tilstra et al., 2017). Over land, the surface albedo is described by a GOME-2 DLER dataset (Tilstra et al., 2019), determined by dividing the GOME-2 orbit swath into five segments and retrieving the traditional surface LER for each segment based on the data from the respective part of orbit swath. Compared to the non-directional GOME-2 LER climatology, the use of the DLER dataset improves the underestimation of the surface albedo at the west side of the GOME-2 orbit (backscattering geometry) and increases the AMFs by up to 15% for polluted regions. Over water, the surface albedo is improved with an ocean surface albedo parametrization (Jin et al., 2011), in which the albedo is parametrized for the direct and diffuse incident radiation separately. We update the simplified expression of diffuse contribution with more realistic values taken from the GOME-2 LER data, and we improve the description of the dependency on viewing direction for the parametrization. The resulting surface albedo increases over sun glint areas and polluted coastal regions with large SZAs and VZAs, for which the tropospheric NO$_2$ columns are reduced by up to 10%.

High-resolution a priori profiles, obtained from the chemistry transport model IFS(CBA) with recent emission inventories, provide a better description of the spatial and temporal variability in the NO$_2$ fields. Compared to the currently used TM5-MP profiles, the application of IFS(CBA) profiles affects the tropospheric NO$_2$ columns by up to $7 \times 10^{15}$ molec/cm$^2$ for polluted regions, mainly due to the differences in the model specifications and model resolutions. To quantify the influence of model resolutions, we implement an analysis by altering the horizontal, vertical, and temporal resolutions for IFS(CBA). Changing the horizontal resolution from 1° to 0.7° affects the tropospheric NO$_2$ columns by up to 20%, with enhanced values for emission sources and the opposite for their unpolluted surroundings. When the vertical resolution changes from 34 layers to 69 layers, the tropospheric NO$_2$ columns increase by up to 15% due to the capture of small box-AMFs at low altitudes. Small differences (< 3%) are found for a temporal resolution of 2-hours and 1-hour time step.

The CAL model from the ROCINN cloud algorithm, with the clouds treated as uniform layers of water droplets, allows the penetration of photons through the clouds and provides more realistic cloud parameters than the current CRB model, with the clouds idealised as Lambertian reflectors. The application of CAL cloud parameters in the AMF calculation takes the sensitivities inside and below the cloud layers into account and reduces the tropospheric NO$_2$ columns by up to $3 \times 10^{15}$ molec/cm$^2$ for polluted regions.

As the cloud retrieval does not distinguish between clouds and aerosols, the aerosol correction is implicitly implemented in the AMF calculation using the cloud parameters. To evaluate the accuracy of the implicit aerosol correction through a cloud model, we explicitly account for the aerosol effect using ground-based aerosol measurements for aerosol-dominated conditions. For Xianghe, a suburban site in China with primarily anthropogenic aerosol emissions, and Bujumbura, a remote site in tropical Africa typically affected by biomass burning aerosols, aerosol optical properties from AERONET measurements and extinction vertical profiles from correlative MAXDOAS measurements are applied. Assuming the explicit aerosol correction as reference, the use of implicit aerosol correction through the CAL cloud model yields a bias 24% smaller than the CRB cloud model for Xianghe and 10% smaller for Bujumbura.



A validation of the improved $NO_2$ measurements is performed by comparing the GOME-2 tropospheric $NO_2$ dataset with
ground-based MAXDOAS measurements at the Xianghe station. The GOME-2 $NO_2$ measurements show similar seasonal
variation as the MAXDOAS dataset with a monthly averaged difference of -9.9% ($-3.8 \times 10^{15}$ molec/cm$^2$ in absolute) and a
correlation coefficient of 0.94, indicating good agreement. The application of the new surface albedo, a priori $NO_2$ profile, and
cloud correction in the AMF calculation improves the biases, correlation coefficients, and regression parameters for Xianghe.

In the future, further studies focusing on the cloud correction will be implemented due to its importance in the AMF cal-
culation. The BRDF effect on cloud parameters will be considered by implementing the GOME-2 DLER dataset in the cloud
retrieval from ROCINN, providing a consistent treatment of surface albedo for both $NO_2$ and cloud retrieval. Note that the
BRDF effect is not discussed for OCRA, because no surface albedo climatology is directly needed, and the correction for
VZA-dependency has been applied in the cloud fraction retrieval as a proxy of BRDF constellation (see Lutz et al., 2016, Sect.
2.2.2 therein). In addition, the interpretation of the OCRA/ROCINN cloud product for aerosol-dominated scenes and the im-
pact on $NO_2$ retrieval algorithm will be further investigated in future studies. Furthermore, the $NO_2$ algorithm will be adapted
to measurements from the TROPOMI instrument with a spatial resolution as high as 7 km$\times$3.5 km.

*Acknowledgements.* This work is funded by the DLR-DAAD Research Fellowships 2015 (57186656) programme with reference number
91585186 and is undertaken in the framework of the EUMETSAT AC-SAF project. We acknowledge the Belgian Science Policy Office
(BELSPO) supporting part of this work through the PRODEX project B-ACSAF. We thank EUMETSAT for the GOME-2 ground segment
interfacing work and for the provision of GOME-2 level 1 products. We thank the UPAS team for the development work on the Universal
Processor for UV/VIS Atmospheric Spectrometers (UPAS) system at DLR. We are thankful to Henk Eskes (KNMI) for creating the TM5-
MP a priori $NO_2$ profiles. We acknowledge the free use of the GOME-2 surface LER database created by KNMI and provided through the
AC-SAF of EUMETSAT. We also acknowledge the free use of the online COART radiative transfer model established at NASA Langley
5  Research Center. We thank the use of imagery from the NASA Worldview application (https://worldview.earthdata.nasa.gov/) as part of the
NASA Earth Observing System Data and Information System (EOSDIS). We also thank the PIs of the AERONET sites used in this study
for maintaining their instruments and providing their data to the community.



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
