# Peer review of "An improved air mass factor calculation for NO2 measurements from GOME-2"

_Atmospheric Measurement Techniques, 2019_

## Referee Comment (RC1) · Anonymous Referee #2 · 10 Sep 2019

The manuscript "An improved air-mass factor calculation for NO$_2$ measurements from GOME-2" by S. Liu et al. describes three improvements to the AMF calculation for GOME-2 NO2 retrievals:

1. Accounting for directional dependence of the surface reflectance

2. Improving the resolution of the NO$_2$ a priori profiles

3. Accounting for clouds in the radiative transfer as layered objects, rather than a single reflective surface.

The work done to implement these improvements is solid. Of the three, the change to how clouds are represented is the most innovative, and while other works have

implemented directionally dependent surface albedo, the authors' approach is distinct from previous approaches and is a clever way of using the GOME-2 instrument directly. These improvements should make the GOME-2 retrieval better, and therefore should be published in AMT after addressing the issues below.

My only major concern with this paper is that the new retrieval is only compared to a single MAX-DOAS site. Given the wealth of data available for NO2 validation (Pandora spectrometers, aircraft data from HIPPO/ATom/DISCOVER-AQ and other campaigns), and that recent validation efforts have seen variation in performance even within one region (e.g. Laughner et al. 2019), a global retrieval really needs to be evaluated in multiple locations around the world. Therefore I strongly recommend that the authors expand the validation before final publication in AMT.

My other general question is relates to the authors' other paper on the GOME-2 retrieval earlier this year (Liu et al. 2019). The authors state that that retrieval is used as the reference retreival for the one presented in this paper. Why are these two upgrades being published separately? Will either be available publicly?

**Specific comments**

- When discussing the $NO_2$ prior profiles, I recommend avoiding characterizing them as "high-resolution", given that $\sim 80$ km is still relatively coarse compared to the horizontal resolution used in regional retrievals (e.g. Russell et al. 2011, McLinden et al. 2014, Goldberg et al. 2017, Laughner et al. 2018).

- Pg. 6, l. 15: Over land, SZA and RAA will be relatively constant at a given latitude over a certain time period, but can vary quite a bit over the course of a year. Have you tested how the DLER varies if you fit the constants in Eq. (4) for different latitude bands and time periods (e.g. 1–3 months)?

- Pg 9: could you be more specific about what characteristics of the DLER are in good agreement with previous work?

- Pg. 15, end of first paragraph: the authors note that the differences in a priori profiles may be due to, among other factors, difference in chemical mechanism? But Table 2 shows no difference in chemical mechanism between the old and new model.

- Related to the previous point: some plots or a discussion of how the emissions differ in the two models would be welcome (perhaps in a supplement).

- Regarding the a priori profiles, how are they matched up to GOME-2 pixels? Interpolation? Area-weighted averaging? Simple averaging? Nearest?

- Pg. 16, l. 8: the authors cite agreement with one study (Kuhlmann et al. 2015) that used much finer horizontal resolution $NO_2$ profiles ($3 \times 3$ km$^2$) than used in this study. Given that other products using $NO_2$ profiles at comparable resolution to the Kuhlmann et al. retreval see larger changes to the $NO_2$ columns (e.g. Russell et al. 2011, McLinden et al. 2014, Goldberg et al. 2017), this comparison needs to be careful to avoid implying that 0.7$°$ resolution for the a priori profiles performs as well as 3 km resolution.

- Pg. 18, end of second paragraph: my understanding is that using the optical centroid as the "cloud top height" was a compromise between simplicity and accounting for the fact that there is penetration of light into the cloud, that the actual top of the cloud is not a hard surface. This line could be written more carefully to acknowledge that.

- Cloud discussion in general: would be helped by making explicitly clear whether the VCDs being considered are total VCDs including the below cloud ghost column via the AMF correction (just to avoid potential confusion).

- Cloud discussion: it might be interesting to discuss how this would affect results from cloud-slicing approaches. Since this shows that the sensitivity to below cloud $NO_2$ is not 0, how should the theoretical framework for cloud slicing be modified?

- Fig. 18 & 19: I don't understand what is meant by "no" aerosol correction. Does this mean that clear-sky box-AMFs were used? If so please state that explicitly, if not, please clarify.

- Table 4: which cloud/aerosol correction is used?

- Tables 4 & 5, Figs. 20 & 21, validation section in general: it would be nice to actually see a comparison of both the operational product and the reference product against the MAX-DOAS data as well as the new product so that we can actually see how the new product is an improvement over those products.

**Technical comments**

- Pg. 3 l. 16: "high horizontal [resolution]" - in the context of NO2 chemistry, 80 km is not really high resolution.

- Table 1: please add to the caption that the different in prior profile models is detailed in table 2.

- Fig. 1c & d: having 0 be at different colors in the two plots is potentially confusing. Recommend standardizing and using the same colormap as in Fig. 3.

- Pg. 10 l. 19: define "case 1 waters"

- Fig 6: does "CIFS" in the legend = IFS(CBA) in the caption, and if so, why are these not the same text?

- Pg. 15: "These values are of the same order of magnitude as the model resolutions of TM5-MP and other chemistry transport models currently employed in the satellite retrieval of NO2" – this is the typical resolution for *global* retrievals, regional retrievals often use 5-80x finer resolution.

- Pg. 15-16: "Consequently, the AMF is underestimated for unpolluted areas and overestimated for polluted areas" - this has been well known for many years (e.g. Heckel et al. 2011, Russell et al. 2011, Valin et al. 2011), please acknowledge previous work.

- Pg. 18, last paragraph: perhaps the discussion of aerosol effects should be moved to the aerosol section? It might make it easier for the reader to follow if this discussion of the shortcomings of the CRB method vis-a-vis aerosols is integrated with the treatment of aerosols in the CAL approach.

**References**

Goldberg et al. A high-resolution and observationally constrained OMI NO$_2$ satellite retrieval. *Atmos. Chem. Phys.*, **17**, 11403–11421, 2017.

Heckel et al. Influence of low spatial resolution a priori data on tropospheric NO$_2$ satellite retrievals. *Atmos. Meas. Tech.*, **4**, 1805–1820, 2011.

Laughner et al. The Berkeley High Resolution Tropospheric NO$_2$ product. *Earth. Sys. Sci. Data*, **10**, 2069–2095, 2018.

Laughner et al. Evaluation of version 3.0B of the BEHR OMI NO$_2$ product. *Atmos. Meas. Tech.*, **12**, 129–146, 2019.

Liu et al. An improved total and tropospheric NO$_2$ column retrieval for GOME-2. *Atmos. Meas. Tech.*, **12**, 1029–1057, 2019.

McLinden et al. Improved satellite retrievals of NO$_2$ and SO$_2$ over the Canadian oil sands and comparisons with surface measurements. *Atmos. Chem. Phys.*, **14**, 3637–3656, 2014.

Russell et al. A high spatial resolution retrieval of NO$_2$ column densities from OMI: method and evalutation. *Atmos. Chem. Phys.*, **11**, 8543–8554, 2011.

Valin et al. Effects of model resolution on the interpretation of satellite NO$_2$ observations. *Atmos. Chem. Phys.*, **11**, 11647–11655, 2011.

---

## Referee Comment (RC2) · Anonymous Referee #1 · 23 Sep 2019

Summary

The authors propose an alternate approach to NO2 retrievals from GOME-2 based on improved, more physically realistic AMF calculations. The reference method assumes clouds are uniform, opaque reflecting boundaries (CRB), assumes climatological isotropic surface reflectivities, and uses a priori NO2 model profiles from TM5. In the new approach, clouds are treated as semi-transparent layers (CAL), the surface reflectivity over land or water has a directional dependence and the model NO2 profiles have been updated. Assumptions regarding aerosols are also explored using MAXDOAS aerosol profiles.

Previous investigations have individually examined some of these model parameters. However this study brings them together to show their combined effects and the net

improvement in the GOME-2 NO2 retrieval. The interdependence of aerosol and cloud-model assumptions is particularly interesting and well-described.

The authors methodically show how the different approaches influence retrieved NO2 and show improvement in correlations with validation data. If the new AMF algorithm can be easily implemented, it represents a very valuable advancement in NO2 retrievals from GOME-2 and potentially other satellite instruments. The paper is extremely well written and organized. I recommend publication after a few minor revisions.

General comments

(1) Please add a short paragraph with general information about the GOME-2 instrument (e.g. launch date, FOV, spectral resolution, etc). Aspects of the AMF modifications have been tested with other satellite instruments, so some context would be useful.

(2) The validations described are very limited, and the GOME-2 footprint is large (40 x 80 km2). It is hard to draw conclusions based on measurements from a single ground-based instrument at a single station, despite the large number of days of data used. The authors allude to the resolution issue by referencing Pinardi et al. (2015), but please add more details about the implications of such limitations. If any additional independent ground-based measurements are available, I strongly recommend including them in the paper.

Specific and minor comments

(1) Page 2, Lines 5 – 8: "...on long time scales. New-generation instruments like the TROPOspheric Monitoring Instrument (TROPOMI) (Veefkind et al., 2012) aboard the Sentinel-5 Precursor satellite and geostationary missions like the Sentinel-4 (Ingmann et al., 2012) will continue this record and deliver NO2 datasets with high spatial resolution and short revisit time."

(2) Page 2, Line 26: "...geometry in the NO2 retrieval (e.g. Boersma..."

(3) Page 2, Line 35: I suggest adding Krotkov et al., 2018 to these citations.

(4) Page 3, Line 8: "...developed and is running operationally..."

(5) Page 3, Line 18: I suggest adding Martin et al., 2002 to these citations regarding cloud influences on the NO2 retrieval:

Martin, R. V., Chance, K., Jacob, D. J., Kurosu, T. P., Spurr, R. J. D., Bucsela, E., Gleason, J., Palmer, P. I., Bey, I., Fiore, A. M., Li, Q., Yantosca, R. M., & Koelmeijer, R. B. A.: An improved retrieval of tropospheric nitrogen dioxide from GOME, J. Geophys. Res., 107(D20), ACH9010ACH9-21, 4437, doi:10.1029/2001JD001027, 2016.

(6) Page 5, Line 3: I suggest Bucsela et al., 2013 as an additional reference on the temperature dependency of the NO2 cross section.:

Bucsela, E. J., Krotkov, N. A., Celarier, E. A., Lamsal, L. N., Swartz, W. H., Bhartia, P. K., Boersma, K. F., Veefkind, J. P., Gleason, J. F., & Pickering, K. E.: A new strato-spheric and tropospheric NO2 retrieval algorithm for nadir-viewing satellite instruments: applications to OMI, Atmos. Meas. Tech.,6, 2607-2626, doi:10.5194/amtd-6-2607-2013, 2013.

(7) Page 5, Line 11: "...cloud-free scenes found with a statistical method..."

(8) Page 6, Line 11: "...from OCRA as inputs."

(9) Page 7, Line 30: "...east of the orbit swath. This effect is larger over snow and ice, due to the forward scattering..."

(10) Page 8, Figure 1: Perhaps the labels over the figures could be written more clearly to show that the first two maps are monthly climatologies, while the others are date-specific.

(11) Page 10, Line 6: "...derives the surface reflectivity for the direct (sun) and diffuse

(sky) incident radiation separately...".

(12) Page 14, Lines 13: "The NO2 data is available on an hourly basis, with profiles at the satellite measurement time obtained by linear interpolation."

(13) Page 15, Lines 8 − 10: "Differences in AMFs are on the order of 0.2 for the Netherlands and 0.02 for China, due to difference in horizontal and vertical resolutions."

(14) Page 19, Lines 6 − 7: "...from an LUT with fixed reference points. This requires no projection from..."

(15) Page 20, Line 11: "...AMF calculation is likely reduced relative to the reference retrieval..."

(16) Page 28, Lines 33 − 34: You may also include Lamsal et al. (2014) in the list of satellite validation studies:

Lamsal, L. N., Krotkov, N. A., Celarier, E. A., Swartz, W. H., Pickering, K. E., Bucsela, E. J., Gleason, J. F., Martin, R. V., Philip, S., Irie, H., Cede, A., Herman, J.,Weinheimer, A., Szykman, J. J., and Knepp, T. N.: Evaluation of OMI operational standard NO2 column retrievals using in situ and surface-based NO2 observations, Atmos. Chem. Phys., 14, 11587–11609, doi:10.5194/acp-14-11587-2014, 2014.

(17) Page 29, Figure 20 and 21 captions: Please explicitly define SAT and GB and state that the light and dark data points in Figure 21 represent daily and monthly-mean differences.

(18) Page 30, Tables 4 and 5: For the uncertainties in MD, please use the standard error of the mean (although you may additionally give the standard deviation).

---

## Author Comment (AC1) · 14 Dec 2019

The comment was uploaded in the form of a supplement:
https://www.atmos-meas-tech-discuss.net/amt-2019-265/amt-2019-265-AC1-supplement.pdf

---

## Author Comment (AC2) · 14 Dec 2019

The comment was uploaded in the form of a supplement:
https://www.atmos-meas-tech-discuss.net/amt-2019-265/amt-2019-265-AC2-supplement.pdf
* * *

---

## Author Response (AR1)

**Response to Referee #1**

We thank the reviewer for the careful reading and the critical remarks, especially about the MAXDOAS validation. Our response to the comments and detailed changes made to the manuscript are described below in black and blue, respectively, and the reviewer's comments are shown in red.

General comments

(1) Please add a short paragraph with general information about the GOME-2 instrument (e.g. launch date, FOV, spectral resolution, etc). Aspects of the AMF modifications have been tested with other satellite instruments, so some context would be useful.

We have extended the introduction of the GOME-2 instrument on Page 2 Line 8:

The GOME-2 instrument, which is the main focus of this study, is included on a series of MetOp satellites as part of the EUMETSAT Polar System (EPS). The first GOME-2 was launched in October 2006 aboard the MetOp-A satellite, and a second GOME-2 was launched in September 2012 aboard MetOp-B. This dataset will be extended with a third GOME-2 aboard the MetOp-C satellite, launched in December 2018. GOME-2 is a nadir-scanning UV-VIS spectrometer measuring the solar irradiance and Earth's backscattered radiance with four main optical channels, covering the spectral range between 240 and 790 nm with a spectral resolution between 0.26 nm and 0.51 nm. The default swath width of GOME-2 is 1920 km, enabling a global coverage in ∼1.5 days. The along-track dimension of the instantaneous field of view is ∼40 km, while the across-track dimension depends on the integration time used for each channel. For the 1920 km swath, the maximum ground pixel size is 80 km×40 km in the forward scan, which remains almost constant over the full swath width. In a tandem operation of MetOp-A and MetOp-B from July 2013 onwards, a decreased swath of 960 km and an increased spatial resolution of 40 km×40 km are employed by GOME-2/MetOp-A (Munro et al., 2016). GOME-2 provides morning observations ...

(2) The validations described are very limited, and the GOME-2 footprint is large (40 × 80 km2). It is hard to draw conclusions based on measurements from a single groundbased instrument at a single station, despite the large number of days of data used. The authors allude to the resolution issue by referencing Pinardi et al. (2015), but please add more details about the implications of such limitations. If any additional independent ground-based measurements are available, I strongly recommend including them in the paper.

Focusing on the available BIRA stations, we have added Tables I to IV in Appendix

B showing the validation results for the urban Uccle station, the remote Bujumbura station, and the background OHP station. As the response to another reviewer, the comparisons are shown for the reference product, the new product, as well as the operational GDP 4.8 product.

The validation of our improved GOME-2 tropospheric $NO_2$ columns is based on ground-based MAXDOAS $NO_2$ measurements at the Xianghe station (39.75° N, 116.96° E) from March 2010 to December 2016, the Uccle station (51° N,4.36° E) from April 2011 to March 2016, the Bujumbura station (3.38° S, 29.38° E) from December 2013 to October 2016, and the OHP station (43.94° N, 5.71° E) from March 2007 to December 2016, respectively. Tables I to IV summarise the validation of the improved retrieval, the reference retrieval, and the current operational GDP 4.8 product against the four MAXDOAS stations. Xianghe is a typical suburban station in northeastern China, Uccle is an urban station in Belgium, Bujumbura is a small city in remote region, and OHP is largely rural but occasionally influenced by polluted air masses transported from neighboring cities.

Table I: Mean difference (MD, SAT-GB in $10^{15}$ molec/cm$^2$), standard deviation (STDEV), standard error of the mean (STERR), relative difference (RD, (SAT-GB)/GB in %), and correlation coefficient R and regression parameters (slope S and intercept I) of the orthogonal regression for the monthly mean GOME-2 tropospheric $NO_2$ product when comparing to MAXDOAS data at the suburban Xianghe station. Values for the improved retrieval (this study) are given, and values for the reference retrieval and the current operational GDP 4.8 product are reported. Results are reported for cloud radiance fraction $< 0.5$ for a total of 77 GOME-2 monthly points.

|  | improved retrieval | reference retrieval | GDP 4.8 product |
| --- | --- | --- | --- |
| MD ($\times 10^{15}$) | -3.8 | -2.7 | -9.2 |
| STDEV ($\times 10^{15}$) | 5.3 | 5.6 | 7.1 |
| STERR ($\times 10^{15}$) | 0.60 | 0.64 | 0.81 |
| RD (%) | -9.9 | -5.8 | -30 |
| R | 0.94 | 0.91 | 0.86 |
| S | 0.69 | 0.73 | 0.63 |
| I | 0.41 | 0.42 | 1.31 |

We have rewritten the introduction and added the implications of the resolution-related issue in Pg 27 L 20:

A validation of our improved GOME-2 tropospheric $NO_2$ columns is performed with BIRA-IASB ground-based MAXDOAS measurements at Xianghe, Uccle, Bujumbura,

Table II: Similar as Table I but for the urban Uccle station for a total of 57 monthly points.

|  | improved retrieval | reference retrieval | GDP 4.8 product |
|---|---|---|---|
| MD ($\times 10^{15}$) | -4.5 | -5.0 | -6.2 |
| STDEV ($\times 10^{15}$) | 2.9 | 2.7 | 3.7 |
| STERR ($\times 10^{15}$) | 0.38 | 0.36 | 0.49 |
| RD (%) | -37 | -43 | -52 |
| R | 0.79 | 0.82 | 0.49 |
| S | 0.53 | 0.47 | 0.35 |
| I | 0.62 | 0.83 | 1.1 |

Table III: Similar as Table I but for the remote Bujumbura station for a total of 36 monthly points.

|  | improved retrieval | reference retrieval | GDP 4.8 product |
|---|---|---|---|
| MD ($\times 10^{15}$) | -3.7 | -3.6 | -3.7 |
| STDEV ($\times 10^{15}$) | 1.9 | 1.8 | 1.1 |
| STERR ($\times 10^{15}$) | 0.32 | 0.30 | 0.18 |
| RD (%) | -76 | -76 | -89 |
| R |  |  |  |
| S |  | na |  |
| I |  |  |  |

Table IV: Similar as Table I but for the background OHP station for a total of 106 monthly points.

|  | improved retrieval | reference retrieval | GDP 4.8 product |
|---|---|---|---|
| MD ($\times 10^{15}$) | -0.82 | -0.85 | -1.2 |
| STDEV ($\times 10^{15}$) | 0.9 | 1.0 | 0.7 |
| STERR ($\times 10^{15}$) | 0.09 | 0.10 | 0.07 |
| RD (%) | -24 | -25 | -45 |
| R | 0.34 | 0.40 | 0.69 |
| S | 0.39 | 0.25 | 0.73 |
| I | 1.1 | 1.2 | -0.5 |

and Observatoire de Haute Provence (OHP), as introduced in Appendix B. The Xianghe station, owing to its polluted suburban nature, is the best site for validation

(Liu et al., 2019). The satellite-based $NO_2$ data is commonly underestimated for urban polluted stations like Uccle, due to the averaging of a local source over a pixel size ($80\times40/40\times40$ km$^2$ for GOME-2) larger than the horizontal sensitivity of the ground-based measurements which is about few to tens of km (Irie et al., 2011; Wagner et al., 2011; Ortega et al., 2015). Difficulties may arise when small local sources are present in remote locations, such as the Bujumbura (urban station in Burundi) or OHP (background station in southern France) sites (Pinardi et al., 2015; Gielen et al., 2017). For the validation of GOME-2 measurements ...

We have extended the analysis in Pg 28 L 10:

Similar figures as Figs. 20 and 21 for the reference retrieval can be found in (Liu et al., 2019), and figures for previous GDP products can be found on the AC-SAF validation website (http://cdop.aeronomie.be/validation/valid-results). Tables I to IV in Appendix B summarize the statistics for the improved retrieval, the reference retrieval, and the operational GDP 4.8 product at Xianghe, Uccle, Bujumbura, and OHP, respectively. As discussed in Pinardi et al. (2015), for remote (Bujumbura) and background (OHP) stations, the mean bias is considered as the best indicator of the validation results, due to the relatively small variability in the measured $NO_2$. In urban (Uccle) and suburban (Xianghe) situations, the $NO_2$ variability is large enough and the correlation coefficient provides a good indication of the coherence of the satellite and ground-based datasets, although larger differences in terms of slope and mean bias can be expected for the urban case, because satellite measurements smooth out the local $NO_2$ hot spots. Comparing to the reference retrieval, the improvement of the algorithm leads to an increase of the correlation coefficient in the suburban Xianghe condition from 0.9 to 0.94 and a decrease of the mean relative bias in the urban Uccle condition from -43% to -37%, and small impacts on the mean bias are found for Bujumbura and OHP. Comparing to the operational GDP 4.8 product, however, both the reference retrieval and the improved retrieval show a significant improvement for all the stations.

As Xianghe is located in a highly polluted region, the presence of large aerosol loads increases the uncertainty on the tropospheric $NO_2$ columns for both the satellite and MAXDOAS retrievals (Richter et al., 2017) and increases the complexity of validation. Since the cloud retrieval can hardly distinguish between clouds and aerosols (see Sect. 4), the "typical" validation results in Table I (cloud radiance fraction < 0.5) do not show improvements e.g. in the mean bias and the slope of the regression line when comparing to the reference retrieval. To separate this effect, the validation at Xianghe is differentiated in completely clear sky in Table 4 below (cloud radiance fraction = 0) and aerosol-dominated cases in Table 5 (cloud radiance fraction < 0.5, cloud optical depth < 5, and cloud top height < 3 km).

Specific and minor comments

(1) Page 2, Lines 5 – 8: "...on long time scales. New-generation instruments like the TROPOspheric Monitoring Instrument (TROPOMI) (Veefkind et al., 2012) aboard the Sentinel-5 Precursor satellite and geostationary missions like the Sentinel-4 (Ingmann et al., 2012) will continue this record and deliver $NO_2$ datasets with high spatial resolution and short revisit time."

Changed.

(2) Page 2, Line 26: "...geometry in the $NO_2$ retrieval (e.g. Boersma..."

Changed.

(3) Page 2, Line 35: I suggest adding Krotkov et al., 2018 to these citations.

Done.

(4) Page 3, Line 8: "...developed and is running operationally..."

Changed.

(5) Page 3, Line 18: I suggest adding Martin et al., 2002 to these citations regarding cloud influences on the $NO_2$ retrieval.

Done.

(6) Page 5, Line 3: I suggest Bucsela et al., 2013 as an additional reference on the temperature dependency of the $NO_2$ cross section.

Done.

(7) Page 5, Line 11: "...cloud-free scenes found with a statistical method..."

Changed.

(8) Page 6, Line 11: "...from OCRA as inputs."

Changed.

(9) Page 7, Line 30: "...east of the orbit swath. This effect is larger over snow and ice, due to the forward scattering..."

Changed.

(10) Page 8, Figure 1: Perhaps the labels over the figures could be written more clearly to show that the first two maps are monthly climatologies, while the others are datespecific.

We have changed the labels to (a) monthly GOME-2 LER climatology and (b) daily improved GOME-2 LER dataset.

(11) Page 10, Line 6: "...derives the surface reflectivity for the direct (sun) and diffuse (sky) incident radiation separately..

Changed.

(12) Page 14, Lines 13: "The $NO_2$ data is available on an hourly basis, with profiles at the satellite measurement time obtained by linear interpolation."

Changed.

(13) Page 15, Lines 8 – 10: "Differences in AMFs are on the order of 0.2 for the Netherlands and 0.02 for China, due to difference in horizontal and vertical resolutions."

As the difference in the vertical resolution affects the example in the Netherlands by 0.03, we would like to keep the general description that the AMFs differ by more than 0.02 for both examples.

(14) Page 19, Lines 6 – 7: "...from an LUT with fixed reference points. This requires no projection from..."

Changed.

(15) Page 20, Line 11: "...AMF calculation is likely reduced relative to the reference retrieval..."

Changed.

(16) Page 28, Lines 33 – 34: You may also include Lamsal et al. (2014) in the list of satellite validation studies.

Done.

(17) Page 29, Figure 20 and 21 captions: Please explicitly define SAT and GB and state that the light and dark data points in Figure 21 represent daily and monthlymean differences.

We have updated the plot legend for Figure 20 and caption for Figure 21.

(18) Page 30, Tables 4 and 5: For the uncertainties in MD, please use the standard error of the mean (although you may additionally give the standard deviation).

Done.

We agree that many additional validation sources are becoming available, and this could be the scope of an additional paper. The present paper mainly illustrates the impact of improving the AMF with DLER, higher temporal and spatial resolution of a priori profiles and a more realistic cloud treatment. Focusing on the available BIRA stations, we have added Tables I to IV in Appendix B showing the validation results for the urban Uccle station, the remote Bujumbura station, and the background OHP station. The comparisons are shown for both the operational GDP 4.8 product and the reference product against the MAXDOAS data as well as the new product.

The validation of our improved GOME-2 tropospheric NO$_2$ columns is based on ground-based MAXDOAS NO$_2$ measurements at the Xianghe station (39.75° N, 116.96° E) from March 2010 to December 2016, the Uccle station (51° N,4.36° E) from April 2011 to March 2016, the Bujumbura station (3.38° S, 29.38° E) from December 2013 to October 2016, and the OHP station (43.94° N, 5.71° E) from March 2007 to December 2016, respectively. Tables I to IV summarise the validation of the improved retrieval, the reference retrieval, and the current operational GDP 4.8 product against the four MAXDOAS stations. Xianghe is a typical suburban station in northeastern China, Uccle is an urban station in Belgium, Bujumbura is a small city in remote region, and OHP is largely rural but occasionally influenced by polluted air masses transported from neighboring cities.

We have rewritten the introduction and added the implications of the resolution-related issue in Pg 27 L 20:

A validation of our improved GOME-2 tropospheric NO$_2$ columns is performed with

Table I: Mean difference (MD, SAT-GB in $10^{15}$ molec/cm$^2$), standard deviation (STDEV), standard error of the mean (STERR), relative difference (RD, (SAT-GB)/GB in %), and correlation coefficient R and regression parameters (slope S and intercept I) of the orthogonal regression for the monthly mean GOME-2 tropospheric NO$_2$ product when comparing to MAXDOAS data at the suburban Xianghe station. Values for the improved retrieval (this study) are given, and values for the reference retrieval and the current operational GDP 4.8 product are reported. Results are reported for cloud radiance fraction < 0.5 for a total of 77 GOME-2 monthly points.

|  | improved retrieval | reference retrieval | GDP 4.8 product |
|---|---|---|---|
| MD ($\times 10^{15}$) | -3.8 | -2.7 | -9.2 |
| STDEV ($\times 10^{15}$) | 5.3 | 5.6 | 7.1 |
| STERR ($\times 10^{15}$) | 0.60 | 0.64 | 0.81 |
| RD (%) | -9.9 | -5.8 | -30 |
| R | 0.94 | 0.91 | 0.86 |
| S | 0.69 | 0.73 | 0.63 |
| I | 0.41 | 0.42 | 1.31 |

Table II: Similar as Table I but for the urban Uccle station for a total of 57 monthly points.

|  | improved retrieval | reference retrieval | GDP 4.8 product |
|---|---|---|---|
| MD ($\times 10^{15}$) | -4.5 | -5.0 | -6.2 |
| STDEV ($\times 10^{15}$) | 2.9 | 2.7 | 3.7 |
| STERR ($\times 10^{15}$) | 0.38 | 0.36 | 0.49 |
| RD (%) | -37 | -43 | -52 |
| R | 0.79 | 0.82 | 0.49 |
| S | 0.53 | 0.47 | 0.35 |
| I | 0.62 | 0.83 | 1.1 |

BIRA-IASB ground-based MAXDOAS measurements at Xianghe, Uccle, Bujumbura, and Observatoire de Haute Provence (OHP), as introduced in Appendix B. The Xianghe station, owing to its polluted suburban nature, is the best site for validation (Liu et al., 2019). The satellite-based NO$_2$ data is commonly underestimated for urban polluted stations like Uccle, due to the averaging of a local source over a pixel size ($80\times40/40\times40$ km$^2$ for GOME-2) larger than the horizontal sensitivity of the ground-based measurements which is about few to tens of km (Irie et al., 2011; Wagner et al., 2011; Ortega et al., 2015). Difficulties may arise when small local sources

Table III: Similar as Table I but for the remote Bujumbura station for a total of 36 monthly points.

|  | improved retrieval | reference retrieval | GDP 4.8 product |
|---|---|---|---|
| MD ($\times 10^{15}$) | -3.7 | -3.6 | -3.7 |
| STDEV ($\times 10^{15}$) | 1.9 | 1.8 | 1.1 |
| STERR ($\times 10^{15}$) | 0.32 | 0.30 | 0.18 |
| RD (%) | -76 | -76 | -89 |
| R |  |  |  |
| S |  | na |  |
| I |  |  |  |

Table IV: Similar as Table I but for the background OHP station for a total of 106 monthly points.

|  | improved retrieval | reference retrieval | GDP 4.8 product |
|---|---|---|---|
| MD ($\times 10^{15}$) | -0.82 | -0.85 | -1.2 |
| STDEV ($\times 10^{15}$) | 0.9 | 1.0 | 0.7 |
| STERR ($\times 10^{15}$) | 0.09 | 0.10 | 0.07 |
| RD (%) | -24 | -25 | -45 |
| R | 0.34 | 0.40 | 0.69 |
| S | 0.39 | 0.25 | 0.73 |
| I | 1.1 | 1.2 | -0.5 |

are present in remote locations, such as the Bujumbura (urban station in Burundi) or OHP (background station in southern France) sites (Pinardi et al., 2015; Gielen et al., 2017). For the validation of GOME-2 measurements ...

We have extended the analysis in Pg 28 L 10:

Similar figures as Figs. 20 and 21 for the reference retrieval can be found in (Liu et al., 2019), and figures for previous GDP products can be found on the AC-SAF validation website (http://cdop.aeronomie.be/validation/valid-results). Tables I to IV in Appendix B summarize the statistics for the improved retrieval, the reference retrieval, and the operational GDP 4.8 product at Xianghe, Uccle, Bujumbura, and OHP, respectively. As discussed in Pinardi et al. (2015), for remote (Bujumbura) and background (OHP) stations, the mean bias is considered as the best indicator of the validation results, due to the relatively small variability in the measured $NO_2$. In urban (Uccle) and suburban (Xianghe) situations, the $NO_2$ variability is large

enough and the correlation coefficient provides a good indication of the coherence of the satellite and ground-based datasets, although larger differences in terms of slope and mean bias can be expected for the urban case, because satellite measurements smooth out the local $NO_2$ hot spots. Comparing to the reference retrieval, the improvement of the algorithm leads to an increase of the correlation coefficient in the suburban Xianghe condition from 0.9 to 0.94 and a decrease of the mean relative bias in the urban Uccle condition from -43% to -37%, and small impacts on the mean bias are found for Bujumbura and OHP. Comparing to the operational GDP 4.8 product, however, both the reference retrieval and the improved retrieval show a significant improvement for all the stations.

As Xianghe is located in a highly polluted region, the presence of large aerosol loads increases the uncertainty on the tropospheric $NO_2$ columns for both the satellite and MAXDOAS retrievals (Richter et al., 2017) and increases the complexity of validation. Since the cloud retrieval can hardly distinguish between clouds and aerosols (see Sect. 4), the "typical" validation results in Table I (cloud radiance fraction < 0.5) do not show improvements e.g. in the mean bias and the slope of the regression line when comparing to the reference retrieval. To separate this effect, the validation at Xianghe is differentiated in completely clear sky in Table 4 below (cloud radiance fraction = 0) and aerosol-dominated cases in Table 5 (cloud radiance fraction < 0.5, cloud optical depth < 5, and cloud top height < 3 km).

My other general question is relates to the authors' other paper on the GOME-2 retrieval earlier this year (Liu et al. 2019). The authors state that that retrieval is used as the reference retreival for the one presented in this paper. Why are these two upgrades being published separately? Will either be available publicly?

The paper Liu et al. 2019 focuses mainly on the improvements to the DOAS retrieval and stratosphere-troposphere-separation, with also an improved AMF calculation based on newer a priori information relative to the current operational GDP product (Valks et al. 2011, 2017). The use of these new features reduces the $NO_2$ column uncertainty and improves the validation results. Since then additional improvements addressing surface albedo, a priori $NO_2$ profile, and cloud and aerosol treatment have been implemented with the aim of further improving the AMF calculation. All these improvements will be implemented in the next version of the GDP.

Specific comments

When discussing the $NO_2$ prior profiles, I recommend avoiding characterizing them as "high-resolution", given that ~80 km is still relatively coarse compared to the horizontal resolution used in regional retrievals (e.g. Russell et al. 2011, McLinden et al. 2014, Goldberg et al. 2017, Laughner et al. 2018).

We have removed the expression "high-resolution" e.g. in Pg 3 L 16, Pg 4 L 20, Pg 16 L 9, and Pg 31 L 19.

Pg. 6, l. 15: Over land, SZA and RAA will be relatively constant at a given latitude over a certain time period, but can vary quite a bit over the course of a year. Have you tested how the DLER varies if you fit the constants in Eq. (4) for different latitude bands and time periods (e.g. 1–3 months)?

The coefficients are fitted for different latitude, longitude, month, and wavelength, as described for Eq. (4). The surface DLER varies with time, and this is partly due to actual changes in the surface properties (with time) and partly due to the changes in the solar angles (with time). The empirical approach that was used to derive the surface DLER does not try to separate these two effects, so it is not possible to extract the impact of the changing solar position from the DLER itself.

We have, however, simulated the impact of the changing solar position on the BRDF/DLER. In Figure I, the change of the SZA over the year is plotted in the top window for a location in the Amazonian rainforest (15°S,65°W). This simulation of the SZA for this location is valid for nadir view geometries (VZA = 0) at a fixed local time of 09:34 LT (corresponding to an hour angle of -36.45 degrees). The twelve horizontal red lines indicate the average SZA of the twelve calendar months.

[Figure]

Figure I: Impact of the changing solar position on the BRDF/DLER over the course of a year at the Amazonian rainforest (15°S,65°W).

The changes in the SZA over the course of a year can be translated into changes in the BRDF using the kernel-based MODIS BRDF approach. The geometric and volumetric kernels (K_geo, K_vol) only need to know the VZA, SZA, and RAA. We use nadir view (VZA = 0) and the SZA as shown in the top window of the figure. The kernel coefficients (f_iso, f_vol, f_geo) are determined from the MODIS MCD43C1 product, for a fixed day (15 March 2008). These kernel coefficients are used for each day of the year, as we are only interested in studying the impact of the change of the solar position.

The change in the BRDF is shown in the middle window of the figure. MODIS band 4 was used, so the relevant wavelength is 555 nm. The twelve horizontal red lines indicate the average BRDF of the twelve calendar months. The difference between the BRDF curve and the red horizontal lines represents the error that is made by assuming that the SZA is more or less constant in the course of a month.

In the bottom window this error in the BRDF is plotted as the green curve. The error in the BRDF is well below 0.001 in almost all cases. If a user of the GOME-2 DLER database applies linear interpolation over time (between the calendar months), then this error should be reduced to the 0.0001 level, as indicated by the blue curve.

Pg 9: could you be more specific about what characteristics of the DLER are in good agreement with previous work?

We have updated the characteristics of the DLER in Pg 9 L 6:

Accounting for the BRDF effect, the surface DLER captures the cross-track dependency of surface albedo, such as the increased reflectivity in backward scattering viewing geometries, in agreement with studies applying the BRDF product from MODIS to describe the dependency of land surface reflectance on illumination and viewing geometry (e.g. Zhou et al., 2010; Noguchi et al., 2014; Vasilkov et al., 2017; Lorente et al., 2018; Laughner et al., 2018; Qin et al., 2019). With a good agreement with the established MODIS BRDF product (Tilstra et al., 2019), both in the absolute sense and in the directional/angular dependency, the GOME-2 DLER dataset ...

Pg. 15, end of first paragraph: the authors note that the differences in a priori profiles may be due to, among other factors, difference in chemical mechanism? But Table 2 shows no difference in chemical mechanism between the old and new model.

We have removed the wrong reason "chemical mechanism" since both of the models apply the same CB05 chemistry.

Related to the previous point: some plots or a discussion of how the emissions differ in the two models would be welcome (perhaps in a supplement).

[Figure]

Figure II: MACCity and CAMS_GLOB_ANT v2.1 anthropogenic surface nitrogen oxides ($NO_x$=$NO_2$+NO) emissions (emitted as nitric oxide, NO) for Europe in August 2010.

We have added the comparisons in Appendix A:

Figure II shows the emission maps of inventories MACCity (Granier et al., 2011) and CAMS_GLOB_ANT v2.1 (Granier et al., 2019) for Europe in August 2010. The annual total anthropogenic emissions are 70.8 Tg NO/yr for MACCity and 73.7 Tg NO/yr for CAMS_GLOB_ANT v2.1. Significant changes in local emission patterns are clear in Fig. II.

Regarding the a priori profiles, how are they matched up to GOME-2 pixels? Interpolation? Area-weighted averaging? Simple averaging? Nearest?

We have added the description in Pg 5 L 20:

... Williams et al. (2013)). The a priori profiles are determined for the GOME-2 overpass time (9:30 LT) and interpolated to the center of the GOME-2 pixel based on four nearest neighbour TM5-MP cell centers.

Pg. 16, l. 8: the authors cite agreement with one study (Kuhlmann et al. 2015) that used much finer horizontal resolution profiles ($3\times3$ km2) than used in this study. Given that other products using $NO_2$ profiles at comparable resolution to the Kuhlmann et al. retreval see larger changes to the $NO_2$ columns (e.g. Russell et al. 2011, McLinden et al. 2014, Goldberg et al. 2017), this comparison needs to be careful to avoid implying that 0.7° resolution for the a priori profiles performs as well as 3 km resolution.

We have removed the misleading comparison and rewritten the finding in Pg 16 L 7:

... 20% for polluted regions, confirming the importance of applying a priori $NO_2$ profiles with better spatial resolution (Heckel et al., 2011; Lin et al., 2014; Kuhlmann et al., 2015).

Pg. 18, end of second paragraph: my understanding is that using the optical centroid as the "cloud top height" was a compromise between simplicity and accounting for the fact that there is penetration of light into the cloud, that the actual top of the cloud is not a hard surface. This line could be written more carefully to acknowledge that.

We have acknowledged this fact by adding in Pg 18 L 33:

... (Saiedy et al., 1967). The retrieved cloud height is normally close to the middle, i.e., the optical centroid of clouds (Ferlay et al., 2010; Richter et al., 2015), which can be considered as a reflectance-weighted height located inside a cloud. Additionally,

since the enhanced multiple scattering is not fully taken into account in the CRB-based cloud retrieval, the retrieved cloud height is close to the altitude of the middle."

Cloud discussion in general: would be helped by making explicitly clear whether the VCDs being considered are total VCDs including the below cloud ghost column via the AMF correction (just to avoid potential confusion).

We have made the fact clear by adding in Pg 6 L 14:

The tropospheric $NO_2$ column calculation is complicated in case of cloudy conditions. For many measurements over cloudy scenes, the cloud top is well above the $NO_2$ pollution in the boundary layer, and the enhanced tropospheric $NO_2$ concentrations cannot be detected by GOME-2 if the clouds are optically thick. Therefore, the tropospheric $NO_2$ column is only calculated for GOME-2 observations with a cloud radiance fraction $\omega < 0.5$. Note that the "below cloud amount" (i.e. the amount of $NO_2$ below the cloud top) for these partly cloudy conditions is implicitly accounted for via the cloudy-sky AMF $M_{cl}$ (in which $m_l = 0$ for layers below the cloud top).

Cloud discussion: it might be interesting to discuss how this would affect results from cloud-slicing approaches. Since this shows that the sensitivity to below cloud $NO_2$ is not 0, how should the theoretical framework for cloud slicing be modified?

The cloud-slicing technique takes advantage of optically thick clouds to estimate the $NO_2$ concentrations in the free troposphere between the clouds. In the cloud-slicing technique, the $NO_2$ profile is assumed to be vertically uniform (not met for highly polluted regions), and thus only pixels with cloud radiance fraction $> 0.9$ is used. For these pixels, the below-cloud contribution to the observed total column is small.

Fig. 18 & 19: I don't understand what is meant by "no" aerosol correction. Does this mean that clear-sky box-AMFs were used? If so please state that explicitly, if not, please clarify.

We have clarified this in Pg 26 L 31:

... no aerosol correction (i.e., applying the clear-sky AMFs) and ...

Table 4: which cloud/aerosol correction is used?

The results are reported for completely clear sky with no aerosol/cloud correction (i.e., applying the clear-sky AMFs), as indicated in the table caption.

Tables 4 & 5, Figs. 20 & 21, validation section in general: it would be nice to

actually see a comparison of both the operational product and the reference product against the MAX-DOAS data as well as the new product so that we can actually see how the new product is an improvement over those products.

We have added the comparisons in Appendix B. Please see response to the general comment.

Technical comments

Pg. 3 l. 16: "high horizontal [resolution]" - in the context of $NO_2$ chemistry, 80 km is not really high resolution.

Changed.

Table 1: please add to the caption that the different in prior profile models is detailed in table 2.

We have added in the caption:

See Table 2 for details on the chemistry transport models used to obtain the a priori $NO_2$ profiles.

Fig. 1c & d: having 0 be at different colors in the two plots is potentially confusing. Recommend standardizing and using the same colormap as in Fig. 3.

We have standardized the colorbar with the white color representing 0 as Fig III.

Pg. 10 l. 19: define "case 1 waters"

We have added the definition in Pg 10 L 19:

open ocean waters dominated by phytoplankton as well as associated products and

Fig 6: does "CIFS" in the legend = IFS(CBA) in the caption, and if so, why are these not the same text?

We have changed to IFS(CBA) for Figs. 6 and 7.

Pg. 15: "These values are of the same order of magnitude as the model resolutions of TM5-MP and other chemistry transport models currently employed in the satellite retrieval of $NO_2$" – this is the typical resolution for global retrievals, regional retrievals often use 5-80x finer resolution.

[Figure]

3 February 2010          5 August 2010

(c) difference over land

surface albedo (improved − original)

-0.02    -0.01    0.00    0.01    0.03    0.04    0.05    0.06

(d) difference over water

surface albedo (improved − original)

0.00    0.003    0.006    0.009    0.011    0.014    0.017    0.02

Figure III: (figure continued from previous page)

We have addressed global in Pg 15 L 5.

Pg. 15-16: "Consequently, the AMF is underestimated for unpolluted areas and overestimated for polluted areas" - this has been well known for many years (e.g. Heckel et al. 2011, Russell et al. 2011, Valin et al. 2011), please acknowledge previous work.

We have added the references (Heckel et al., 2011; Russell et al., 2011; Valin et al., 2011) in Pg 16 L 2.

Pg. 18, last paragraph: perhaps the discussion of aerosol effects should be moved to the aerosol section? It might make it easier for the reader to follow if this discussion of the shortcomings of the CRB method vis-a-vis aerosols is integrated with the treatment of aerosols in the CAL approach.

As Fig. 10 shows the differences in cloud top heights obtained with the CRB and CAL models, followed by an analysis of the distinct opposite (positive and negative)

difference for different aerosol cover. This effect is also repeated in the aerosol section like Fig. 16. We have added a reference in Pg 19 L 11:

[revised manuscript text omitted]